# Pericyte remodeling is deficient in the aged brain and contributes to impaired capillary flow and structure

Andrée-Anne Berthiaume[1,2], Franca Schmid [3,4,10], Stefan Stamenkovic[1,10], Vanessa Coelho-Santos[1,10], Cara D. Nielson [1,5], Bruno Weber [4,6], Mark W. Majesky[1,7] & Andy Y. Shih [1,2,8,9] ✉

Deterioration of brain capillary flow and architecture is a hallmark of aging and dementia. It remains unclear how loss of brain pericytes in these conditions contributes to capillary dysfunction. Here, we conduct cause-and-effect studies by optically ablating pericytes in adult and aged mice in vivo. Focal pericyte loss induces capillary dilation without blood-brain barrier disruption. These abnormal dilations are exacerbated in the aged brain, and result in increased flow heterogeneity in capillary networks. A subset of affected capillaries experience reduced perfusion due to flow steal. Some capillaries stall in flow and regress, leading to loss of capillary connectivity. Remodeling of neighboring pericytes restores endothelial coverage and vascular tone within days. Pericyte remodeling is slower in the aged brain, resulting in regions of persistent capillary dilation. These findings link pericyte loss to disruption of capillary flow and structure. They also identify pericyte remodeling as a therapeutic target to preserve capillary flow dynamics.

Alterations in brain capillary architecture commonly occur with aging. Loss of capillary density, increased tortuosity and twisting of vessels, formation of non-patent string capillaries, and abnormal lumen diameter are consistent observations across decades of neuropathological research on the aged brain[1]. When these defects accumulate in the capillary network, they hinder the distribution of oxygen-carrying blood cells, leading to tissue hypometabolism and heightened sensitivity to disease processes. In Vascular Contributions to Cognitive Impairment and Dementia (VCID) and Alzheimer's disease (AD), these capillary defects are seen at higher frequencies and contribute to neurodegeneration[1]. Despite a wealth of knowledge on their association with cognitive decline, the etiology of age-related capillary changes remains poorly understood, due in part to the cross-sectional

nature of human autopsy studies and limited resolution of in vivo clinical imaging. Lacking mechanistic insight, therapeutic access points to mitigate age-related capillary insufficiency are missing.

Pericytes are mural cells that line the capillary beds of the brain[2]. They serve essential roles in the formation and maturation of blood vessels[3], establishment of blood-brain barrier (BBB) integrity[4,5], and regulation of capillary diameter and blood flow[6,7]. Recent studies suggest that pericytes are also sensitive to neuronal activity and participate in neurovascular coupling[8,9]. Pericytes form a near-continuous cellular chain along the entire capillary network in healthy microvasculature, leaving little of the capillary bed uncontacted by their extensive processes. Each pericyte occupies a distinct territory, which does not overlap with the territories of its immediate neighbors[10,11].

[1]Center for Developmental Biology and Regenerative Medicine, Seattle Children's Research Institute, Seattle, WA, USA. [2]Department of Neuroscience, Medical University of South Carolina, Charleston, SC, USA. [3]Institute of Fluid Dynamics, ETH Zurich, Sonneggstrasse 3, Zurich, Switzerland. [4]Institute of Pharmacology and Toxicology, University of Zurich, Winterthurerstrasse 190, Zurich, Switzerland. [5]Graduate Program in Neuroscience, University of Washington, Seattle, WA, USA. [6]Neuroscience Center Zurich, University and ETH Zurich, Winterthurerstrasse 190, Zurich, Switzerland. [7]Department of Laboratory Medicine and Pathology, University of Washington, Seattle, WA, USA. [8]Department of Pediatrics, University of Washington, Seattle, WA, USA. [9]Department of Bioengineering, University of Washington, Seattle, WA, USA. [10]These authors contributed equally: Franca Schmid, Stefan Stamenkovic, Vanessa Coelho-Santos. ✉e-mail: Andy.Shih@SeattleChildrens.org

Mounting evidence indicates that pericytes are hyper-sensitive to cerebrovascular pathology, and loss of pericyte number and endothelial coverage occurs in normal aging[12] and is accelerated in both VCID and AD[13–17]. Pericyte loss has been predominantly linked to deficits in BBB integrity[13,17,18]. However, it is unclear how this loss is related to deficiencies in capillary structure and perfusion in the aged and diseased brain[19].

In prior work, we demonstrated the necessity of pericyte coverage to maintain capillary tone and flow resistance in vivo[6]. Optical ablation of individual capillary pericytes led to aberrant dilation of uncovered capillaries and increased blood flow. This pericyte loss also triggered the growth of processes from neighboring pericytes to re-establish endothelial contact and restore vascular tone, revealing an intrinsic repair mechanism[10]. These studies raised several questions. First, the consequence of focal pericyte loss on broader capillary flow dynamics remained unknown. Second, it was unclear if structural remodeling was the primary means of restoring endothelial coverage with greater degrees of pericyte loss. Third, the capacity for pericyte remodeling was not examined in the aging brain, the context in which pericyte remodeling is most important. Here, we address these questions in brain capillary networks of healthy adult and aged mice using in vivo two-photon single cell ablation and longitudinal imaging.

## Results

### No difference in capillary landscape in upper cortex of adult and aged mice

We compared capillary structure and pericyte abundance at baseline between adult and aged PDGFRβ-tdTomato mice using in vivo two-photon imaging (Fig. 1a,b, Supplementary Fig. 1a,b). Our analyses focused on capillary networks within the first 10–100 μm from the cortical pial surface, as subsequent pericyte ablation studies were limited to these depths. Capillary networks of aged mice (18–24 months) did not differ in total length, number of junctions, or abundance of pericyte somata when compared to adult mice (3–6 months)(Supplementary Fig. 1c, d, e). Further, the limited capillary regions that lacked pericyte coverage did not differ between ages (Supplementary Fig. 1f). Thus, capillary structure and pericyte coverage in upper layers of cortex are comparable at baseline between adult and aged mice.

### Acute consequence of triple pericyte ablation

In vivo optical ablation of capillary pericytes allowed us to observe the response to focal pericyte loss. We ablated three contiguous pericytes from capillary regions in upper layers of cortex in adult and aged mice (Fig. 1c-f). Across both age groups, triple pericyte ablations uncovered capillary lengths of 643.75 ± 26.43 μm (mean ± SEM; 475–809 μm range)(Fig. 1d,g), which encompassed ~15 total capillary segments, i.e., sections between branch-points (Fig. 1h). The affected individual capillary segments varied greatly in length (4.79–236.6 μm)(Fig. 1i). Triple pericyte ablation resulted in loss of pericyte–pericyte contact at 7–10 process termini from an average of eight neighboring pericytes, each with potential to grow into the uncovered territory (Fig. 1d,j,k). We detected no difference in these basal vascular or pericyte metrics between adult and aged mice.

### Pericyte remodeling is efficient in the adult mouse brain, but impaired with aging

To measure pericyte remodeling, we re-examined capillary regions with pericyte loss at 3, 7, 14, and 21 days post-ablation (Fig. 2). In adult mice, neighboring pericytes remodeled their processes to achieve contact with nearly the entire uncovered region within 21 days (Fig. 2a,b). In a representative example, endothelial contact was regained through simultaneous growth of ten neighboring pericytes, each extending a single process toward the uncovered region (Fig. 2c). Remodeling led to eventual contact with another pericyte process

terminus, i.e., pericyte-pericyte contact, upon which no further growth was possible (Fig. 2d)[10]. In the aged group, a similar number of remodeling processes contributed to endothelial re-coverage, but a greater amount of capillary length remained uncovered at 21 days (Fig. 2e–h).

In the adult brain, 84% of processes had made pericyte-pericyte contact within the 21-day timeframe (Fig. 3a,b). In the aged brain, less than 40% of processes had made contact. To assess the dynamics of pericyte process growth, we measured process length and growth rate for each interval of imaging, i.e., day 3 to 7 (Fig. 3c,d). "Maximum growth rate" was the time interval with the fastest growth, whereas "average growth rate" was the mean rate across all time intervals before pericyte-pericyte contact was achieved. We observed a significant decrease in both maximum and average growth rates in aged mice (Fig. 3e,f). After 21 days, however, the final length of extension achieved and total process length (from soma to process terminus) was not statistically different between age groups (Fig. 3g,h). An age-related effect in these metrics may be masked by the high variance of remodeling between pericytes, and because pericytes were studied broadly across the arterio-venous axis, as will be discussed below.

To better understand the outcome of synergistic remodeling, we calculated the length of endothelium uncovered over time. In adult mice, uncovered capillary length declined steeply in the first 7 days following ablation, and then slowly approached zero over the next 14 days (Fig. 3i, left). The aged group displayed a slower decline of uncovered territory, with a portion of the endothelium still left uncovered after 21 days (Fig. 3i, right). In a subset of experiments in aged mice, ablation areas were re-visited 49 days post-ablation and sizeable stretches of capillary remained uncovered (Fig. 3j, Supplementary Fig. 2). Together, these data show that the interconnected structure of the capillary network facilitates restoration of endothelial coverage, but the capacity for pericyte remodeling is reduced with aging.

Since tdTomato expression was driven under constitutive pdgfrb promotor activity, new pericytes arising from proliferation or migration would be fluorescently labeled. However, we did not observe the emergence of new pericyte somata in either age group, suggesting that remodeling of existing pericytes is the main reparative strategy at this scale of pericyte loss.

### Contributors to the heterogeneity of pericyte process growth

We explored what parameters might contribute to variability in pericyte process growth. Many remodeling processes met one or more capillary bifurcations as they entered the uncovered territory, when the processes would split and extend in both directions, unless the terminus of another pericyte impeded its growth (Supplementary Fig. 3a). A single process could split up to three times, resulting in four growing termini. Given this parallel growth, processes that split had significantly greater final extension lengths than those that did not (Supplementary Fig. 3b). In adult mice, 49% of processes split, as opposed to 35% in aged mice, a difference attributed to their slower growth rather than differences in the number of capillary bifurcations available (Supplementary Fig. 1d). There was no difference in the number of bifurcations within uncovered regions between age groups (7.3 ± 2.7 and 7.9 ± 2.9 bifurcations uncovered in adult and aged mice, respectively; $p$ = 0.4035, $t$-test; mean ± SEM).

We further examined if the maximum process length achieved was limited by baseline process length. Baseline lengths did not differ between age groups, consistent with normal pericyte density at baseline (Supplementary Fig. 3c). We, therefore, pooled age groups and plotted baseline process length as a function of length extended post-ablation. This revealed a negative correlation suggesting limited growth potential in individual processes longer than ~150 μm (Supplementary Fig. 3d). It also showed that short basal processes that split during growth had the greatest final added length.

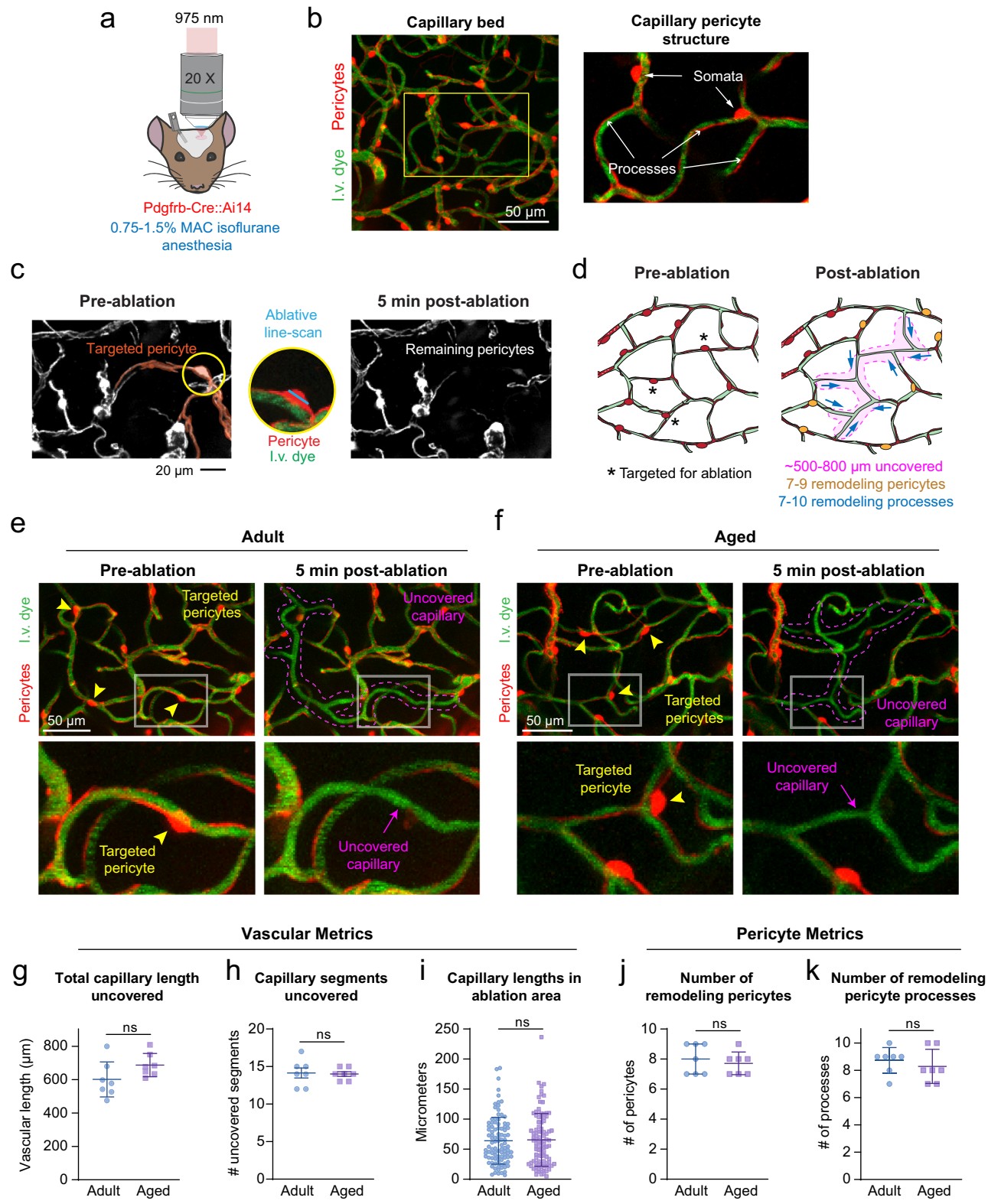

In both age groups, ~25% growing pericyte processes created small vessel distortions ("microbends") when they tethered to a point further along the capillary and pulled the anchored point closer (Supplementary Fig. 4). Microbends were transient and capillary shape returned to normal once the process terminus had extended further. Processes that created microbends grew at a faster rate and gained more contact distance (Supplementary Fig. 4e,f). Thus, both intrinsic pericyte properties (baseline process length, ability to distort vessel)

and extrinsic factors (capillary branching) can influence pericyte growth potential.

## Mural cells across microvascular zones can contribute to capillary re-coverage

The cortical microvasculature is divided into arteriole-to-capillary transition, capillary, and venular zones, across which mural cells vary in their transcriptional profile, morphology, and contractile dynamics[20,21].

**Fig. 1 | Acute consequence of ablating multiple contiguous pericytes. a** In vivo two-photon imaging through a chronic cranial window in anesthetized PDGFRβ-tdTomato mice. **b** Capillary bed with vessels labeled green with intravenous (i.v.) dye, FITC-dextran (70 kDa), and pericytes genetically labeled in red. The inset shows the structure of capillary pericytes. Representative of 12 PDGFRβ-tdTomato mice in this study. **c** A capillary pericyte targeted for two-photon ablation encircled in yellow. Ablative line-scanning is restricted to the pericyte soma. Post-ablation image shows loss of fluorescence from targeted pericyte, including its processes, within minutes post-ablation. Neighboring pericytes are unaffected. Representative of >42 pericyte ablations. **d** Schematic showing metrics collected following ablation of three contiguous pericytes. Left, targeted pericytes are marked with asterisks. Right, metrics include total capillary length uncovered, number of remodeling pericytes, and number of remodeling pericyte processes. **e,f** Before and 5 min after triple pericyte ablation in adult and aged animal. Arrowheads indicate targeted pericyte somata, and dotted magenta line outlines approximate regions of

uncovered endothelium. Inset shows a magnified region with pericyte ablation and uncovered territory 5 min afterwards. Each example is representative of 7 experiments per age group. Comparison of vascular metrics between adult and aged mice after triple pericyte ablation. **g** Total capillary length uncovered; $t(12) = 1.790$, $p = 0.0988$. **h** Number of fully and partially uncovered capillary segments; $t(12) = 0.1936$, $p = 0.8497$. **i** Range of capillary segment lengths; $t(195) = 0.2325$, $p = 0.8164$. All are unpaired $t$ tests (two-sided), for $n = 7$ triple pericyte ablations (6 adult mice), 7 triple pericyte ablations (6 aged mice) for **g** and **h** and $n = 99$ capillary segments over 6 adult mice, $n = 98$ capillary segments over 6 aged mice for **i**. Data are shown as mean ± SD. Ns = non-significant. Comparison of pericyte metrics between adult and aged mice after triple pericyte ablation. **j** Number of remodeling pericytes; $t(12) = 0.6030$, p=0.5577. **k** Number of remodeling pericyte processes; $t(12) = 0.7206$, $p = 0.4850$. Unpaired $t$ tests for $n = 7$ triple pericyte ablations (6 adult mice), 7 triple pericyte ablations (6 aged mice). Data are shown as mean ± SD.

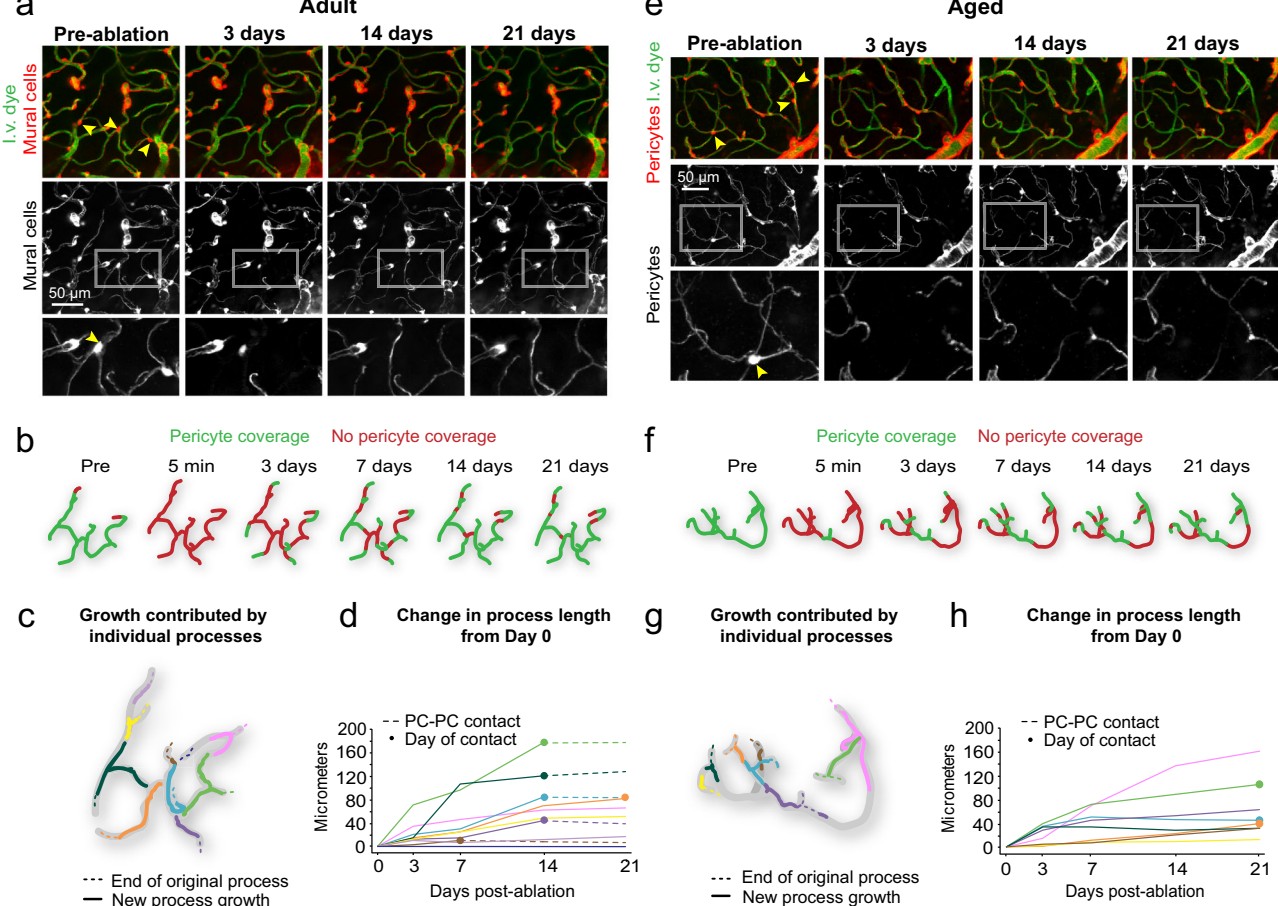

**Fig. 2 | Pericyte remodeling ensures capillary coverage following pericyte loss. a** Representative example of pericyte remodeling from an adult mouse (4 months old at pre-ablation), showing the area before triple pericyte ablation, and at select time-points after ablation. Arrowheads indicate pericytes targeted for ablation. The lower panels show mural cells only. Insets show magnified view of region with pericyte ablation followed with remodeling of neighboring pericyte processes. Each example is selected from 7 experiments in adult mice. I.v. dye = intravenous dye. **b** Schematic of capillary coverage by pericytes on each imaging day. **c** Schematic representation of growth contributed by different pericyte processes

over 21 days. Each color represents an individual pericyte process, with location of original terminus of the process as a dashed line. **d** Plot of added pericyte process length over time from Day 0. Each color corresponds to an individual pericyte process from **c**. Closed circles mark the day of pericyte-pericyte contact, after which growth stops (dashed lines). PC-PC contact = pericyte–pericyte contact. **e–h** Representative example from an aged mouse (21 months old at pre-ablation), with same layout as for adult mouse. Each example is selected from 7 experiments in aged mice.

Although pericytes were exclusively ablated in the capillary zone, they were often adjacent to arteriole–capillary transition and venular zones, allowing us to assess remodeling of mural cells from these regions into the capillary zone (Fig. 4a).

Mural cells from all zones were able to extend processes into the uncovered capillary territory. Interestingly, regardless of the

morphological characteristics of their existing processes, mural cells extended into the capillary zone with a "thin-strand" morphology typical of capillary pericytes (Fig. 4b–d, Supplementary Fig. 5a–d). Striking examples are ensheathing pericytes and venular smooth muscle cells (venule SMCs), whose pre-existing processes were more complex (Fig. 4b,d) This suggests that pericyte process morphology

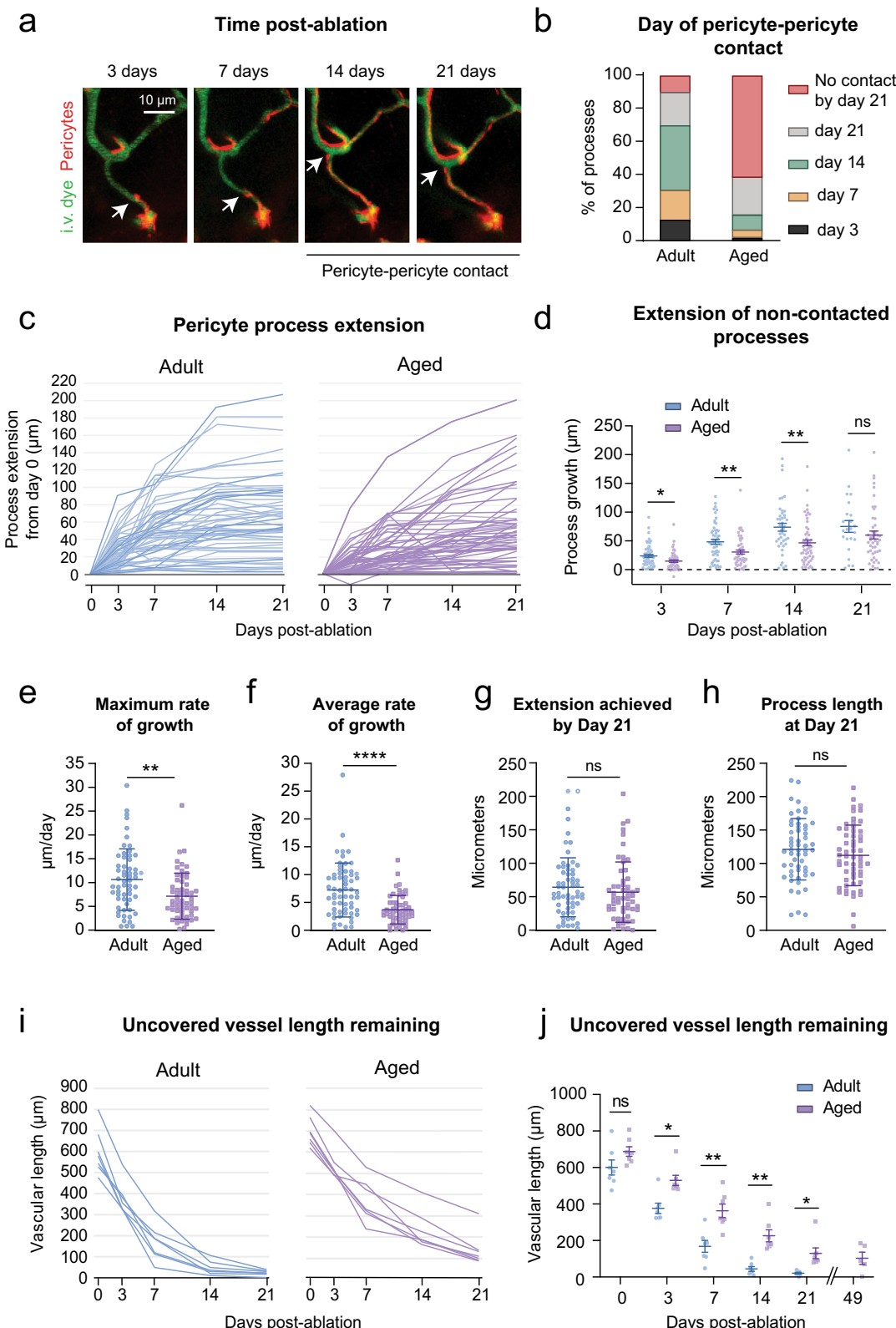

is dictated by the endothelial zone it contacts. In further support of this concept, we also ablated ensheathing pericytes and found that remodeling neighbors retained their circumferentially-oriented processes when extending within the arteriole-capillary transition zone (Supplementary Fig. 5e,f).

We found no difference in rate of process growth across micro-vascular zones in adult mice (Fig. 4e). The aged group exhibited

selective impairment in remodeling of ensheathing and capillary pericytes (mesh and thin-strand morphology) of the arteriole-capillary transition and capillary zones, respectively. In contrast, venule SMCs on ascending venules maintained their remodeling capacity (Fig. 4e, Supplementary Fig. 5c,d). Comparing maximal process extension between adult and aged mice for each vascular zone yielded a similar outcome (Fig. 4f). This points to venule SMCs as a retained source of

**Fig. 3 | Inefficient pericyte remodeling in the aged brain fails to regain complete capillary coverage. a** In vivo two-photon images of a pericyte process growing over time (white arrow), until it is inhibited by pericyte–pericyte contact. This is a representative example of pericyte-pericyte contact from 21 pericyte ablations. I.v. dye = intravenous dye. **b** Percent of total pericyte processes making pericyte-pericyte contact on each imaging day across age groups. $n = 60$ processes from 6 adult mice, $n = 57$ processes from 6 aged mice. **c** Pericyte process extension over time in each age group, including all processes examined. $n = 60$ processes from 6 adult mice, $n = 57$ processes from 6 aged mice. **d** Plot showing the extension of pericyte processes, excluding those that have made pericyte-pericyte contact, at each post-ablation imaging day. Mixed-effects analysis with Sidak's multiple comparisons (two-sided), $F_{(1,115)} = 9.979$; overall effect **$p = 0.002$, Day 3, *$p = 0.0187$; Day 7, **$p = 0.0056$; Day 14, **$p = 0.0087$; Day 21, $p = 0.6574$. $N = 60$ processes from 6 adult mice, and $n = 58$ processes from 6 aged mice. Data are shown as mean ± SEM. **e, f** Maximum and average rate of pericyte process growth in aged compared to adult mice. **e** Unpaired $t$ tests (two-sided) with Welch's correction for unequal variances: $t_{(109.1)} = 3.313$; **$p = 0.0013$; **f** $t_{(90.50)} = 4.977$; ****$p < 0.0001$. $N = 60$

processes from 6 adult mice, $n = 57$ processes from 6 aged mice. Data are shown as mean ± SD. **g** Maximum process extension achieved in 21 days. Unpaired $t$ test (two-sided), $t_{(115)} = 0.9370$; $p = 0.3507$. $n = 60$ processes from 6 adult mice, 57 processes from 6 aged mice. Data are shown as mean ± SD. **h** Total process length at day 21. $t_{(112)} = 1.066$; $p = 0.2889$. Data in **e–h** shown as mean ± SD. Statistics performed with unpaired t tests (two-sided) for $n = 60$ processes from 6 adult mice, $n = 57$ processes from 6 aged mice. Data shown as mean ± SD. **i** Uncovered vessel length remaining over time. Each line represents one triple pericyte ablation region. Adult, $n = 7$ triple pericyte ablations; 6 mice; Aged $n = 7$ triple pericyte ablations from 6 mice. **j** Comparison of average vessel length remaining on each imaging day across age groups. Two-way ANOVA with Sidak's multiple comparisons test, $F_{(1,12)} = 16.72$; overall effect **$p = 0.0015$, Day 0, $p = 0.4196$. Day 3, *$p = 0.0118$. Day 7, **$p = 0.0098$. Day 14, **$p = 0.0052$. Day 21, *$p = 0.0490$. Adult, $n = 7$ regions from 6 mice; aged, $n = 7$ regions from 6 mice. Day 49 is from a subset of triple pericyte ablation experiments in aged animals only (5 regions from 4 mice), not included in statistical analyses. Data are shown as mean ± SEM.

repair capacity within the aged brain and suggests that arteriole-capillary transition and capillary zones may be more susceptible to prolonged loss of pericyte coverage.

### Overt BBB leakage is very rare following focal pericyte ablation

Pericytes are well-established as custodians of BBB integrity[5]. In recent studies, BBB permeability occurring from adult-induced pericyte loss could be detected by extravasation of a 70 kDa dextran dye[22] and even larger IgG proteins[23]. In both adult and aged mice, we detected no extravasation of 70 kDa dye after focal loss of pericyte coverage loss (Supplementary Fig. 6a,b). However, in a rare occurrence in an aged mouse (1 out of 7 triple ablation experiments; 1 of 98 uncovered capillary segments inspected) leakage was observed on days 3 and 7 post-ablation (Supplementary Fig. 6c). We also assessed a low molecular weight dextran dye in aged mice (10 kDa FITC-dextran), which stays within the blood plasma for several minutes following a single i.v. bolus. No increase in dye extravasation was seen after pericyte loss (3 days post-ablation), compared to regions receiving off-target sham irradiations where the ablative laser path was placed away from a pericyte soma, but of similar distance to the vessel wall (Fig. 5a–d).

We considered the possibility that BBB leakage was too slow or subtle to be detected by imaging of dye extravasation. Since microglia are highly sensitive to vascular pathology[24,25], cluster around leaky vessels[26], and react to BBB leakage in models with more extensive pericyte loss[22], we reasoned that they would be sensitive indicators of any BBB leakage. In adult and aged double transgenic mice with co-labeled mural cells and microglia, microglia reacted within minutes to engulf the soma of the ablated pericyte (Fig. 5e,g,i). Critically, this reaction was wholly focused on the soma of the dying pericyte, and not the broader capillary regions previously contacted by their processes. Following off-target sham irradiations, microglial processes occasionally extended to investigate the area in the 30 min following laser exposure, but this reaction was comparatively mild (Fig. 5f,h,j). Three days post-ablation or off-target irradiation, we observed no aggregation of microglia or their processes along peri-vascular regions suggesting no delayed BBB leakage.

We further examined endothelial cell junctions after focal pericyte ablation. Using Claudin5-eGFP fusion protein mice, prior studies showed the development of protrusions and small gaps within tight junction strands during stroke-induced BBB leakage, indicative of endothelial remodeling and BBB damage, respectively[27]. After pericyte ablation in adult mice with co-labeled pericytes and tight junctions, we observed a transient but non-significant increase in small protrusions, and no increase in tight junction gaps (Supplementary Fig. 7). We also used adeno-associated virus to label astrocytes in aged mice. No overt changes in astrocyte endfoot apposition to the capillary wall following focal pericyte ablation were seen (Supplementary Fig. 8). Thus, the

degree of pericyte loss in our paradigm does not reach the threshold needed to cause overt BBB disruption, except with one rare observation in an aged mouse.

### Aberrant capillary dilation with loss of pericyte coverage is exacerbated with aging

In prior studies, we reported that loss of pericyte coverage led to abnormal capillary dilation in adult mice, indicating a role in regulation of basal capillary tone[6,10]. We compared this effect between adult and aged mice with triple pericyte ablations. At baseline, we detected no difference in the diameter of capillaries across age groups (Fig. 6a). For each ablation experiment, we measured the diameters of 4–5 capillaries at the following time-points: i) Pre-ablation, when the capillary was covered, ii) 3 days post-ablation, when the endothelium was uncovered, and iii) 14–21 days post-ablation, when the endothelium was re-covered by process growth from neighboring pericytes (Fig. 6b). In adult mice, we observed average dilations of 0.6 μm (21% increase from baseline) when the capillary was uncovered (Fig. 6c,d,g). Dilation occurred exclusively where pericyte contact was lacking (Supplementary Fig. 9). With coverage regained by pericyte remodeling, capillary diameters returned to levels no different from baseline (Fig. 6h).

In aged mice, uncovered capillary segments dilated to a significantly greater extent than in adult mice, i.e., an average of 1.1 μm (36% increase from baseline), with some individual capillaries dilating as much as -1.6 μm (Fig. 6e–g). Further, slight dilations persisted even after pericyte coverage had returned (Fig. 6h), suggesting inefficient re-establishment of vascular tone. Capillaries that continued to lack coverage beyond 21 days post-ablation maintained their dilated states (Fig. 6i). We confirmed that the dilations were not due to the laser damage alone, as off-target sham irradiations led to no change in capillary diameter (Supplementary Fig. 10). Further, these results were not an effect of isoflurane anesthesia, as dilations persisted when mice were imaged in the awake state (Supplementary Fig. 11). Altogether, this indicates that pericyte contact is key to maintaining basal capillary tone in vivo, and dilation with pericyte loss is exacerbated in the aged brain.

### Local dilations alter flow distribution and increase flow heterogeneity in capillary networks

The flow and oxygenation of red blood cells (RBCs) in brain capillaries is heterogeneous at rest[6,28]. Homogenization of blood flow among capillaries is necessary to efficiently extract and distribute oxygen ($O_2$) to the tissue[29–32]. Therefore, increased flow heterogeneity is a barrier to achieving flux homogenization and $O_2$ extraction. To examine whether focal capillary dilations increased flow heterogeneity, we performed triple pericyte ablations (Fig. 7a)

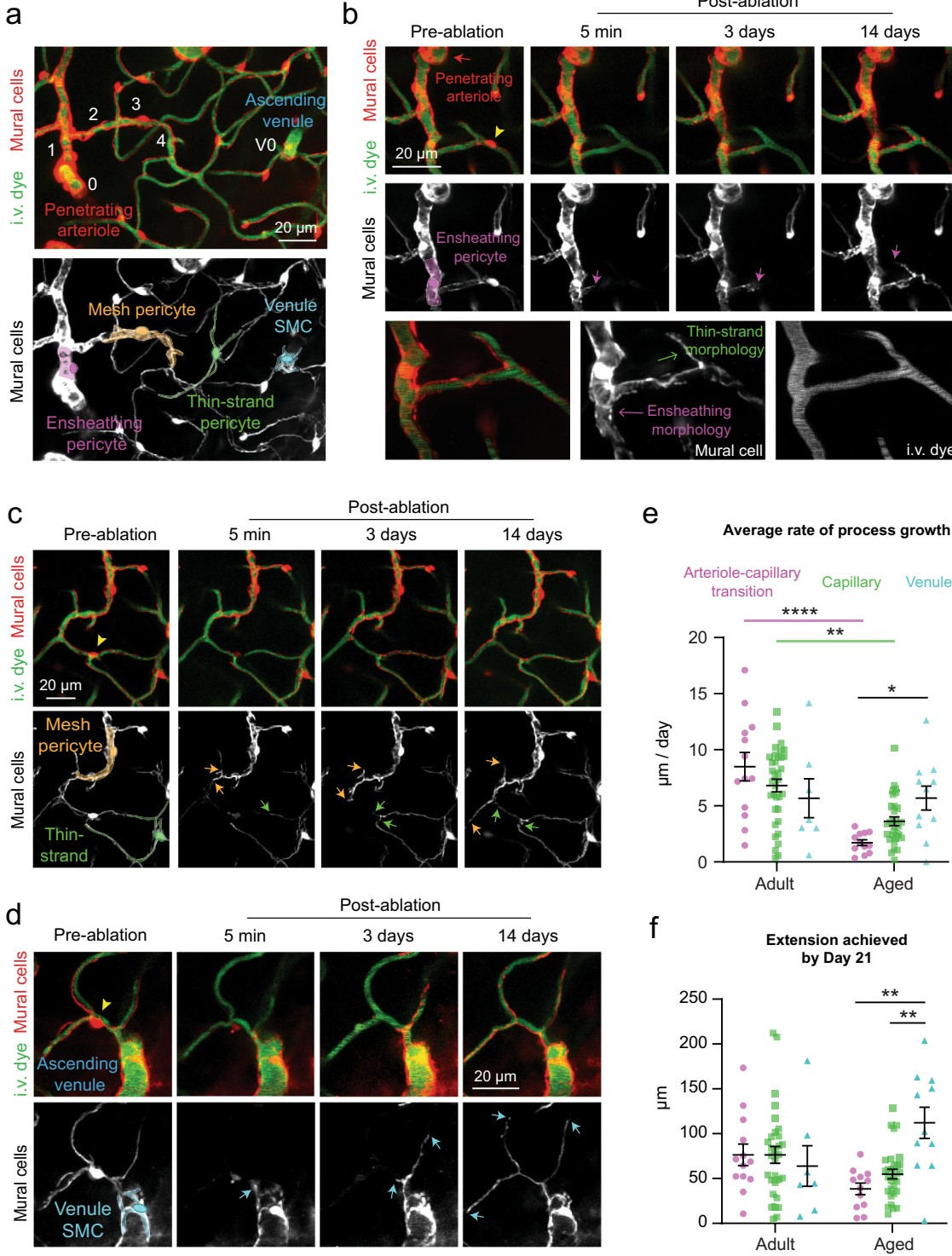

and used line scans to measure blood cell flux (cells/s) and flow directionality in capillaries both within and immediately surrounding the uncovered region (Fig. 7b,c). Flux was measured in the same capillaries before and 3 days post-ablation, when pericyte remodeling had not yet restored coverage and tone.

In an aged mouse, the already high variance of basal flux among capillaries, 18 to 231 cells/s, further broadened to 37 to 453 cells/s post-ablation (Fig. 7d). The increased flux was predominantly observed in dilated capillary segments lacking pericyte coverage (Fig. 7b,e,f). However, a fraction of sampled capillaries surrounding these segments decreased in flux (Fig. 7b,e,g). When pericyte

contact was lost in only one of the daughter branches of a divergent bifurcation, blood cells flowed preferentially through the dilated branch. This left the alternate branch under-perfused relative to baseline, even when overall flow entering the bifurcation was higher post-ablation (Fig. 7h-j). This non-uniform partitioning of blood cell flow at capillary bifurcations reflects the Zweifach-Fung effect[33], where increased flow rate in a dilated capillary pulls a larger fraction of RBCs into the dilated daughter branch. These local upstream changes can have propagating effects within the capillary network[34], as we also observed decreased flow at downstream, convergent bifurcations (Fig. 7k–m).

**Fig. 4 | Mural cell remodeling capacity varies with age across microvascular zones. a** In vivo two-photon image of a microvascular network from penetrating arteriole to ascending venule, with branch orders in white. Bottom image highlights mural cell subtypes within zones. This is a representative view from 14 separate ablation experiments in regions containing different microvascular zones. I.v. dye = intravenous dye. Venule SMC = venular smooth muscle cell. **b** Example of an ensheathing pericyte from an adult mouse remodeling into the capillary zone. Arrow indicates position of leading process terminus over time. Bottom row shows high-resolution image of the cell at 21 days post-ablation. This is a representative example from 13 processes observed in arteriole-capillary transition zone. **c** Example of a mesh and thin-strand pericyte from an adult mouse growing within the capillary bed. Arrows show growing terminal ends of processes. This is a representative example from 35 processes observed in capillary zone. **d** Example of a venule SMC process from an adult mouse growing into the capillary bed, as indicated by arrows. This is a representative example from 7 processes observed in venular zone. **e** Average process growth by mural cells in different microvascular zones. Two-way ANOVA with Tukey's multiple comparisons test (two-sided), For age comparison, $F_{(1,102)} = 21.45$; overall effect, ****$p = 0.0001$; Arteriole-capillary transition (adult vs. aged) ****$p < 0.0001$; capillary (adult vs. aged) **$p = 0.0016$;

Venule (adult vs. aged) $p > 0.9999$. For vessel type comparison, $F_{(2,102)} = 0.1838$; overall effect, $p = 0.8324$, For interaction between age and vessel type, $F_{(2,102)} = 6.029$, overall effect, **$p = 0.0033$. Arteriole-capillary transition vs. capillary (adult), $p = 0.5833$; arteriole-capillary transition vs. venule (adult), $p = 0.4125$; capillary vs. venule (adult), $p = 0.9533$. Arteriole-capillary transition vs. capillary (aged), $p = 0.4855$; arteriole-capillary transition vs. venule (aged), *$p < 0.0398$; capillary vs. venule (aged), $p = 0.4516$. Adult: $n = 13$ arteriole-capillary transition, $n = 35$ capillary, $n = 7$ venule from 6 mice; aged: $n = 12$ arteriole-capillary transition, $n = 30$ capillary, $n = 10$ venule from 6 mice. Data are shown as mean ± SEM. **f** Maximum process extension by mural cells in different microvascular zones. Two-way ANOVA. For age comparison, $F_{(1,97)} = 0.1455$; overall effect, $p = 0.7037$. For vessel type comparison, $F_{(2,97)} = 2.613$; overall effect, $p = 0.0784$. For interaction between age and vessel type, $F_{(2,97)} = 5.532$; **$p = 0.0053$. Arteriole-capillary transition vs. capillary (adult), $p > 0.9999$; arteriole-capillary transition vs. venule (adult), $p < 0.9897$; capillary vs. venule (adult), $p = 0.3511$. Arteriole-capillary transition vs. capillary (aged), $p = 0.8750$; arteriole-capillary transition vs. venule (aged), **$p < 0.0014$; capillary vs. venule (aged), **$p = 0.0044$. Adult: $n = 13$ arteriole-capillary transition, $n = 35$ capillary, $n = 7$ venule from 6 mice; aged: $n = 12$ arteriole-capillary transition, $n = 30$ capillary, $n = 10$ venule from 6 mice. Data are shown as mean ± SEM.

## Focal capillary dilation increases flow heterogeneity in silico

To understand capillary flow changes on a broader network level, we performed blood flow simulations in realistic microvascular networks derived from mouse parietal cortex (Fig. 8a)[35]. Four cases were studied in two microvascular networks, each involving dilation of 13–14 contiguous capillary segments with similar characteristics to those targeted during triple pericyte ablations in vivo (Fig. 8b)(Supplementary Table 1). The capillaries were each dilated by 0.6 μm or 1.1 μm from their baseline diameters to mimic average dilations measured in the adult and aging brain, respectively (Fig. 6g). An additional group with 1.6 μm dilation was included to examine the effect of the most extreme dilations observed in aged mice. Consistent with observations in vivo, focal dilations produced both increases and decreases in capillary flow (defined as >10% change from baseline)(Fig. 8c). Flow perturbations were detected in hundreds of capillaries surrounding the dilated region. The number of affected capillaries increased commensurately with the extent of dilation, with ~1/3 of the affected capillaries decreasing in flow irrespective of the extent of dilation. Capillaries with increased flow localized generally to the epicenter of dilation, while those with decreased flow resided in surrounding areas (Fig. 8d).

To understand how capillary architecture related to blood flow change, we categorized capillaries into three groups: Dilated, Gen1-Neighbors (1 branch from dilated), and Gen2-Neighbors (2 branches from a dilated segment). Consistent with in vivo observations, dilated capillaries generally increased in flow, while Gen1 and Gen2 neighbors showed both increases and decreases from baseline (Fig. 8e). Further, capillary steal was observed at divergent capillary bifurcations, where one daughter branch was selectively dilated and the undilated branch experienced decreased flow (Fig. 8f). As predicted by the Zweifach-Fung effect[33], the steal effect is even more pronounced with RBC flux (Supplementary Fig. 12a). Consistent with in vivo data, the variance in absolute capillary flow increased substantially with focal capillary dilation (Supplementary Fig. 12b). To assess whether capillary flow heterogeneity deviated from a normal range, we defined the expected baseline flow heterogeneity among a multitude of undilated capillaries within the two microvascular networks (Fig. 8g). This revealed that capillary dilation in the adult brain led to abnormally elevated regional flow heterogeneity, and that heterogeneity further increases with magnitudes of dilation seen in the aged brain (Fig. 8h).

## Local dilations produce blood flow stalls and capillary regression in vivo

In surveying volumetric data collected across all in vivo triple ablation experiments, we found a higher frequency of blood flow stalls in experiments involving pericyte ablation, compared to off-target irradiations (Fig. 9a-d, Supplementary Fig. 13). These stalls appeared as

vessels with no moving blood cell shadows within the dye-labeled plasma, and could be detected in images without line-scanning[36]. Stalls occurred with greatest likelihood one branch-point away from an uncovered capillary segment (Fig. 9e), suggesting blood steal by an adjacent dilated capillary. Many of the stalls occurred at divergent bifurcations, which occur closer to the arteriole-capillary transition zone (Supplementary Fig. 13b).

Most capillary stalls were transient, and presumably resolved by restoration of capillary tone in dilated neighbors. Transient stalls were defined as capillary segments with no blood flow on one day of imaging, but with flow observed in a subsequent imaging session. We could not determine the precise duration over which flow was stalled, due to limited sampling frequency. However, we suspect that some capillaries experienced prolonged loss of flow, as a subset of stalling events were later associated with regression of the non-flowing capillary segment (13% of total stalls). Regressions left behind a pericyte bridge without an endothelial lumen, reminiscent of a string capillary (Fig. 9b,d,f, Supplementary Fig. 13c). These outcomes were based on pooling data across adult and aged mice, as their frequency in focal ablation experiments was insufficient for age comparison. Additionally, we did not detect increases in stalled or low-flow capillaries in the in silico data (Supplementary Fig. 12c,d). This is not surprising since biological processes such as blood cell adhesion to the endothelium and potential vasoconstrictive events were not captured in our modeling. Collectively, these data reveal how altered capillary flow distribution due to pericyte loss can lead to cessation of blood flow in capillary segments, and lead to enduring loss of capillary structure.

## Discussion

We have ablated pericytes focally in capillary networks of the adult and aged mouse brain to understand how pericyte loss affects capillary function in vivo, and to study pericyte remodeling in the context of brain aging. Our experiments show that restoration of endothelial coverage following pericyte loss is a synergistic effort from multiple neighboring pericytes, leveraging highly interconnected capillary networks to create access points for remodeling pericytes. This suggests that pericyte remodeling would be less efficient in aging or disease, particularly in sparser capillary networks of white matter and hippocampus[37–39]. Resilience would be further diminished in the arteriole-capillary transition and capillary zones of the aged brain due to impaired pericyte growth.

Abnormal capillary dilations caused by pericyte loss lead to marked increases in local blood flow[6]. However, uneven dilation among capillaries results in steal of flow from capillary segments due to abnormal partitioning of blood cell flux at bifurcations. This finding is supported by concordant data from in silico and in vivo experiments

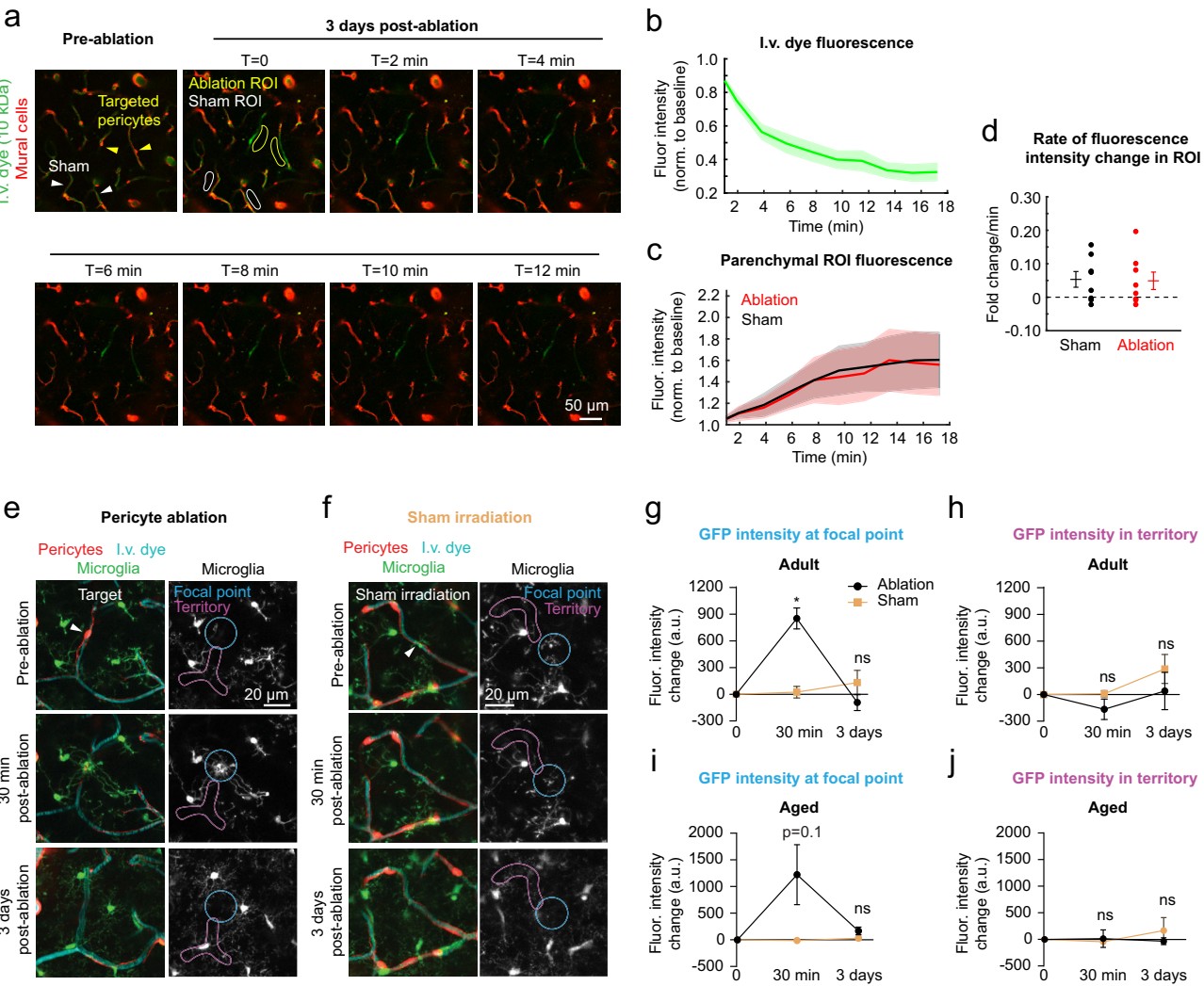

**Fig. 5 | Lack of overt BBB disruption or perivascular inflammation with focal pericyte loss. a** Imaging time course to examine for extravasation of intravenous (i.v.) 10 kDa FITC-dextran dye in aged mice. Pre-ablation image shows location of targeted pericytes and sham irradiation controls. At day 3, T=0 shows regions of interest (ROI) from which fluorescence intensity measurements were collected over time. This is a representative example from 8 ablation experiments. **b** Intensity of FITC-dextran fluorescence within the capillary lumen as a function of time post-injection. Data shown as mean ± SEM. Fluor. Intensity = fluorescence intensity. **c** Intensity of FITC-dextran fluorescence in parenchymal ROI between ablation and sham regions. Data are shown as mean ± SEM. **d** Rate of fluorescence intensity change between ablation and sham ROI regions. Wilcoxon rank sum test (two-sided), $p = 0.904$. $N = 7$ ablation regions and $n = 7$ sham regions from 4 aged mice. Data shown as mean ± SEM. In vivo two-photon images from adult Pdgfrβ-tdtomato;Cx3Cr1-GFP mice, showing microglia reaction to **e** pericyte ablation or **f** sham

irradiation after 30 min and 3 days. ROIs are drawn in the region directly exposed to laser ablation or irradiation (focal point), or the capillary segments covered by pericyte processes (territory). This is a representative example from 3 ablation experiments in adult mice. GFP intensity in focal point (**g**) and territory (**h**) after pericyte ablation and sham in adult mice, shown in arbitrary units (a.u.). Two-way repeated measures ANOVA with Sidak's multiple comparisons (two-sided). Focal point: $F_{(1.45, 7.252)} = 24.41$, overall effect ***$p = 0.0009$. Pericyte ablation vs. sham irradiation *$p = 0.0216$ at 30 min; $p = 0.5504$ at 3 days. Territory: $F_{(1.149, 5.744)} = 3.170$, $p = 0.1263$. GFP intensity in focal point (**i**) and territory (**j**) in aged mice. Two-way repeated measures ANOVA. Focal point: $F_{(1.03, 7.183)} = 3.214$, overall effect $p = 0.1146$. Territory: $F_{(1.761, 12.33)} = 0.2922$, $p = 0.7248$. For **g** and **h**, $n = 3$ pericyte ablations, 4 sham irradiations from 5 mice between 3 and 6 months of age. For **i** and **j**, $n = 5$ pericyte ablations, 4 sham irradiations from 3 mice between 18 and 24 months of age. Data presented as mean ± SEM.

showing reduction of flow in ~1/3 of affected capillaries, and increased flow heterogeneity. The brain's ability to homogenize flux among capillaries is critical for proper allocation of $O_2$ to brain tissue[30,31]. Broadening of capillary flux heterogeneity by pericyte loss creates a larger barrier for flux homogenization, limiting tissue oxygenation at rest and during functional hyperemia. Indeed, greater capillary flow heterogeneity has been associated with age-related cerebrovascular diseases underlying stroke and dementia[40,41]. In future studies, it will be important to examine the consequence of these capillary abnormalities on tissue $O_2$ content[42].

Pericyte loss also led to increased blood flow stalling. Stalls were observed in more than half of our experiments, and we estimate that an

average of 3%, and maximum of 9%, of capillaries in the region (including dilated vessels, Gen1 and Gen2 neighbors) experienced a stall in flow. Flow stalls are typically seen in only 0.4% of brain capillaries of healthy adult mice, and an increase to 1.8% in a mouse model of AD was associated with worsened cognitive performance[36]. In agreement with this idea, a recent study using an adult-induced genetic pericyte depletion model also observed increased capillary stalling[43]. This effect was attributed to enhanced leukocyte–endothelial interaction, but may also involve flow steal in capillary networks.

Capillary networks of adult mouse cortex are also remarkably stable in structure[44]. Our observation of capillary regression with ablations, but not shams, suggests that flow disruptions due to

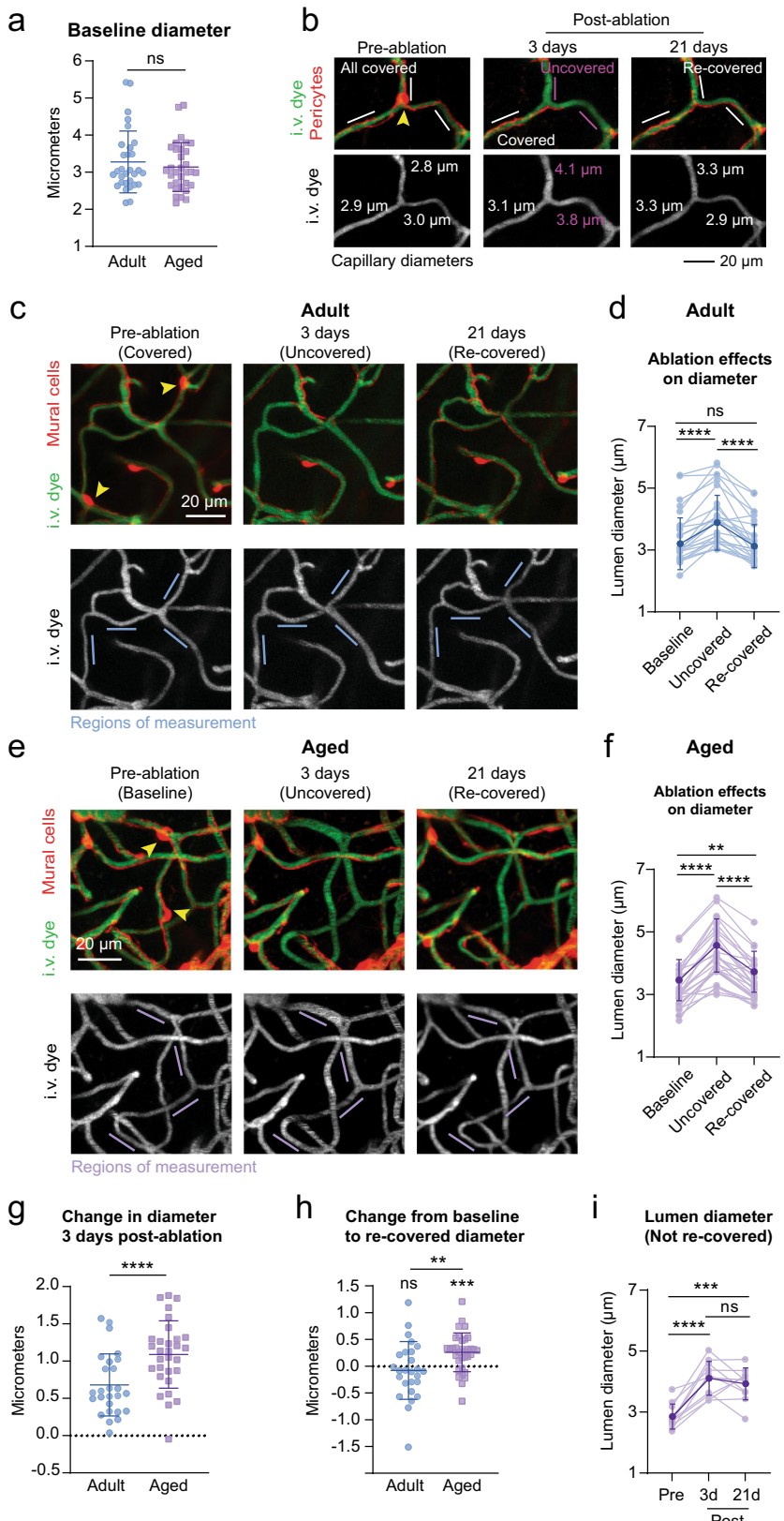

pericyte loss can promote enduring reductions in capillary connectivity. Collectively, these data establish a link between pericyte loss and cerebral blood flow deficiency, wherein aberrant network flow dynamics are a key facet of the impairment. The effects we report here are based on a restricted scale of pericyte loss, and we expect capillary defects to compound with the severity of pericyte loss reported in AD

(30–60% loss of total pericyte numbers)[13,16]. Further, we suspect that string capillaries and overall reduced capillary density seen in aging and dementia may result, in part, from capillary flow abnormalities induced by pericyte loss[19,45].

Our findings help to explain previously observed changes in capillary structure and function with brain aging. While we did not

**Fig. 6 | Pericyte coverage loss leads to augmented capillary dilations in the aged brain. a** Comparison of baseline capillary diameters across age. Unpaired $t$ test (two-sided), $t(60) = 0.7245$; $p = 0.4716$. $N = 30$ capillaries from 6 adult mice, $n = 32$ capillaries from 6 aged mice. Data shown as mean ± SD. Ns = non-significant. **b** Capillary dilation occurs in region with loss of pericyte coverage, but not in adjacent capillaries that maintain pericyte coverage. Arrowhead points to pericyte targeted for ablation. I.v. dye = intravenous dye. **c** Example capillary region from adult mouse following triple pericyte ablation. Arrowheads show two targeted cells. Bottom row shows i.v. dye alone, with capillary segments measured over time identified by blue lines. **d** Plot of capillary diameters over time from adult mice. Repeated measures one-way ANOVA with Tukey's multiple comparisons test (two-sided), $F(1.824, 45.60) = 35.95$, ****$p < 0.0001$. Baseline vs uncovered, ****$p < 0.0001$; baseline vs. re-covered, $p = 0.7393$; uncovered vs. re-covered, ****$p < 0.0001$. $N = 26$ capillaries from 6 mice. Data are shown as mean ± SD. **e** Example capillary region from aged mouse following triple pericyte ablation. Arrowheads point to targeted pericytes. Bottom row shows i.v. dye alone, with capillary segments measured over time identified by purple lines. **f** Plot of capillary diameters over time from aged mice. Repeated measures one-way ANOVA with Tukey's multiple comparisons test (two-sided), $F(1.914, 53.59) = 111.3$, ****$p < 0.000$. Baseline vs uncovered, ****$p < 0.0001$; baseline vs. re-covered, **$p = 0.0018$; uncovered vs. re-covered, ****$p < 0.0001$. $N = 29$ capillaries from 6 mice. Data shown as mean ± SD. **g** Change in diameter from baseline at 3 days post ablation, when capillaries lack pericyte coverage. Unpaired $t$ test (two-sided), $t(54) = 3.501$, ***$p = 0.0009$. $N = 26$ capillaries in 6 adult mice, $n = 30$ capillaries in 6 aged mice. Data are shown as mean ± SD. **h** Difference between baseline and re-covered capillary diameter detected in adult and aged groups. Adult: One sample $t$ test (two-sided), $t(25) = 0.7451$; $p = 0.4631$, for $n = 26$ from 6 mice. Aged: One sample $t$ test (two-sided), $t(33) = 4.172$; ***$p = 0.0002$, for $n = 34$ capillaries from 6 mice. Difference in re-covered diameter between adult and aged mice. Unpaired $t$ test with Welch's correction (two-sided), $t(41.40) = 2.751$; **$p = 0.0088$. Data shown as mean ± SD. **i** Dilations persist for capillaries that do no regain pericyte coverage at 21 days in aged mice. Repeated measures one-way ANOVA with Tukey's multiple comparisons test (two-sided), $F(1.967, 19.67) = 31.35$; overall effect ****$p < 0.0001$. Pre-ablation vs 3 days, ****$p < 0.0001$; pre-ablation vs 21 days, ***$p = 0.0002$; 3 days vs 21 days, $p = 0.5976$. $N = 11$ capillaries from 6 mice. Data are shown as mean ± SD.

detect basal differences in capillary density or pericyte abundance in superficial somatosensory cortex, broader examinations by other groups have reported lower capillary density in cortical gray and white matter[37,38,46], and reduction in both pericyte number and coverage[12] in the aged rodent brain. Two-photon imaging studies in aged rats[47] and mice[48] have revealed average increases in capillary diameter and RBC velocity, which in combination with reduced capillary density and hematocrit, become associated with microscopic pockets of under-perfused tissues[48]. Recent microvascular angiography studies in the aged mouse brain have also revealed greater heterogeneity in flow velocity across cortical capillaries under basal conditions[49]. It will be important to establish how spontaneous, age-related pericyte loss affects capillary perfusion and structure, particularly in the white matter where capillary rarefaction in aging is widely observed.

Studies using global pericyte depletion have shown marked decreases in neurovascular coupling responses[50,51], BBB disruption and circulatory failure due to vasogenic edema[23]. Our studies provide a clearer view on how pericyte loss affects capillary flow, since the focal ablation approach affects blood flow without disrupting the BBB or inducing overt vascular pathology. However, the lack of BBB leakage in our model does not contradict prior work showing BBB leakage with more extensive pericyte loss[22,23]. It is likely that pericytes use paracrine signaling pathways to ensure overall endothelial health, and ablation of a few isolated pericytes does not compromise overall barrier function.

PDGF-B/PDGFRβ signaling is involved in pericyte coverage during adulthood since genetic ablation of endothelial PDGF-BB in adult mice leads to age-dependent pericyte loss and BBB defects[22]. Augmenting PDGF-B/PDGFRβ activity in adulthood may therefore be a strategy to increase pericyte coverage and improve capillary homeostasis. In vitro, administration of PDGF-BB leads to proliferation of human brain pericytes and protection from apoptosis[52]. Intravenous treatment with PDGF-BB peptide was sufficient to promote pericyte coverage and restore capillary tone and flow rate in a model of epilepsy involving mural cell loss[53]. Interestingly, the immunomodulatory drug, thalidomide, can increase expression of PDGF-BB from endothelial cells, promoting pericyte coverage and reducing bleeding in models exhibiting vascular pathology in the CNS[54] and peripheral tissues[55,56]. While deeper mechanistic studies are needed, these studies form the logic to develop therapeutics that can promote pericyte coverage and preserve normal capillary flow dynamics in age-related cerebrovascular diseases.

## Methods
### Animals
PDGFRβ-tdTomato mice[57] were bred from Pdgfrb-Cre (FVB/NJ background)[58] and Ai14 (C57BL/6 background)[59] mice and aged in-house for the study. The adult mouse group ($n = 6$; 3 female) included animals between 3 to 6 months of age. The aged mouse group ($n = 6$; 3 female) included animals between 18 and 24 months of age. Age group brackets were determined by the Jax "Life Span as a Biomarker" criteria, which defines 3–6 months as "mature adult" stage, and 18–24 months as "old" stage in mice. Aged mice were visually checked for general health and any overt signs of illness such as tumors or weight loss resulted in exclusion from the study. For a subset of experiments, we crossed PDGFRβ-tdTomato mice with CX3CR1-GFP (C57BL/6 background)[60,61] or Claudin-eGFP fusion protein (C57BL/6 background)[27] mice to simultaneously label mural cells alongside microglia or endothelial tight junctions, respectively. PDGFRβ-tdTomato;CX3CR1 mice were used at 3–6 months of age (adult; $n = 5$; 2 female) and 18–24 months of age (aged; $n = 5$; 1 female) before the start of experiments. PDGFRβ-tdTomato;Claudin-eGFP mice ($n = 3$; 1 female) were aged to 7–12 months. The blood cell flux example (Fig. 7) is from a 19-month-old male PDGFRβ-tdTomato mouse. Mice were maintained on a 12-h light cycle (7:00am on, 7:00 pm off). Room temperature and humidity were maintained within 68–79 °F (setpoint 73 °F) and 30–70% (setpoint 50%), respectively. Mouse chow (LabDiet PicoLab 5053 irradiated diet for standard mice, and LabDiet PicoLab 5058 irradiated diet for breeders) was provided ad libitum. The Institutional Animal Care and Use Committee at the Seattle Children's Research Institute approved all procedures used in this study (protocol IACUC00419).

### Surgery
Chronic cranial windows (skull removed, dura intact) were implanted in the skulls of all mice for triple pericyte ablation experiments. Briefly, surgical plane anesthesia was induced with a cocktail consisting of fentanyl citrate (0.05 mg/kg), midazolam (5 mg/kg) and dexmedetomidine hydrochloride (0.5 mg/kg)(all from Patterson Veterinary). Dexamethasone (40 μL; Patterson Veterinary) was given 4–6 h prior to surgery to reduce brain swelling during the craniotomy. Circular craniotomies ~3 mm in diameter were generated under sterile conditions, and sealed with a glass coverslip consisting of a round 3 mm glass coverslip (Warner Instruments; 64-0720 (CS-3R)) glued to a round 4 mm coverslip (Warner Instruments; 64-0724 (CS-4R)) with UV-cured optical glue (Norland Products; 7110). The coverslip was positioned with the 3 mm side placed directly over the craniotomy, while the 4 mm coverslip laid on the skull surface at the edges of the craniotomy. An instant adhesive (Loctite Instant Adhesive 495) was carefully dispensed along the edge of the 4 mm coverslip to secure it to the skull. Lastly, the area around the cranial window was sealed with dental cement. This two-coverslip "plug" fits precisely into the craniotomy and helps to inhibit skull regrowth, thereby preserving the optical clarity of the window over months. Mice recovered for a minimum of 3 weeks following surgery.

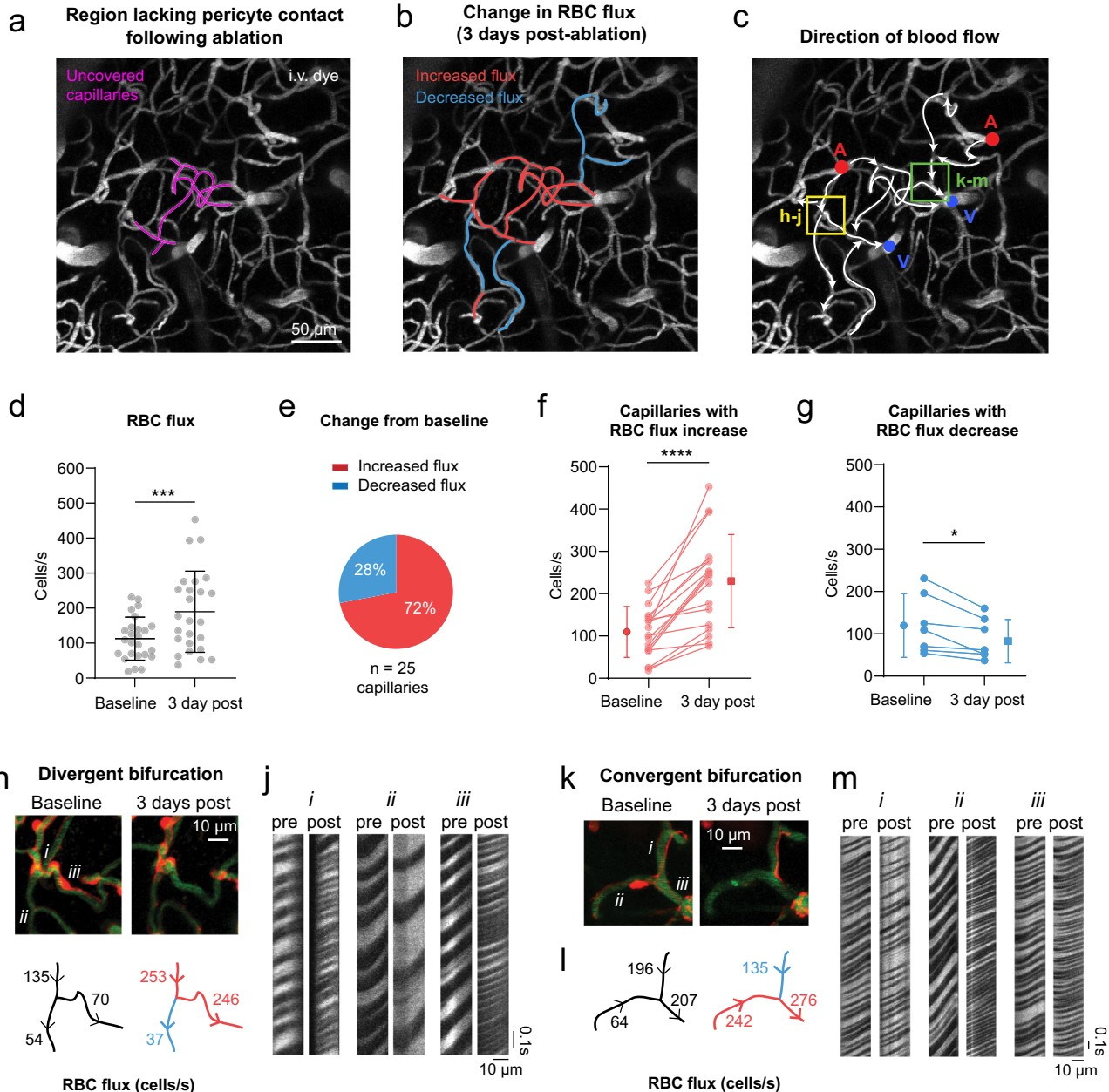

**Fig. 7 | Blood flow changes in regions of pericyte ablation and surrounding capillaries. a** Image showing capillaries uncovered by a triple ablation (purple). These data are from a single triple pericyte ablation experiment, performed in an aged mouse. I.v. dye = intravenous dye. **b** Three days post-ablation, changes in red blood cell (RBC) flux from baseline are color-coded in a subset of vessels. **c** Overlay of blood flow direction on vascular image. Ascending venules are marked in blue, and penetrating arterioles in red. **d** Blood cell flux before and 3 days post ablation for segments in and immediately around the ablation area. Paired $t$ test, $t(24)$ =3.880; ***$p$ = 0.0007 for $n$ = 25 capillaries from an aged mouse. Significantly higher variance was detected at 3 days compared to baseline, F test to compare variances (two-sided), $F(24, 24)$ = 3.545; ***$p$ = 0.0029. Data shown as mean ± SD. **e** Proportion of vessels in the area that experienced an increase vs. decrease in flux on Day 3. **f** Blood cell flux in capillaries that increased in flux from baseline to 3 days post-ablation. Paired $t$ test (two-sided), $t(17)$ = 6.273; ****$p$ < 0.0001, for $n$ = 18 capillaries from an aged mouse. **g** A subset of capillaries decreased in flux following pericyte ablation. Paired $t$ test (two-sided), $t(6)$ = 3.229; *$p$ = 0.0179, $n$ = 6 capillaries from an aged mouse. Data in **d**, **f**, **g** are shown as mean ± SD. **h–j** A divergent bifurcation that loses pericyte coverage in one downstream branch, while the alternate route remains covered. RBCs passing through i.v. dye captured by line scans are shown for each vessel (i, ii, iii) over time. **k–m** A convergent bifurcation that loses pericyte coverage to one upstream vessel. RBCs passing through i.v. dye captured by line scans are shown for each vessel (i, ii, iii) over time.

In a subset of experiments investigating the response of microglia to the pericyte ablation, thinned-skull windows (PoRTs window technique) were used to minimize disruption of the brain and best preserve resting state microglia[62]. This window involved using a dental drill to thin the skull for optical penetrance, but without breaching the calvaria. These windows were imaged one day following surgery for a maximum of three days.

**Two-photon imaging**

In vivo two-photon imaging system was performed with a Bruker Investigator (run by Prairie View 5.5 software) coupled to a Spectra-Physics Insight X3. Green, red and far red fluorescence emission was collected through 515/30 nm, 615/60 nm, 700/75 nm bandpass filters, respectively, and detected by gallium arsenide phosphide photomultiplier tubes. Low-resolution maps of the cranial window

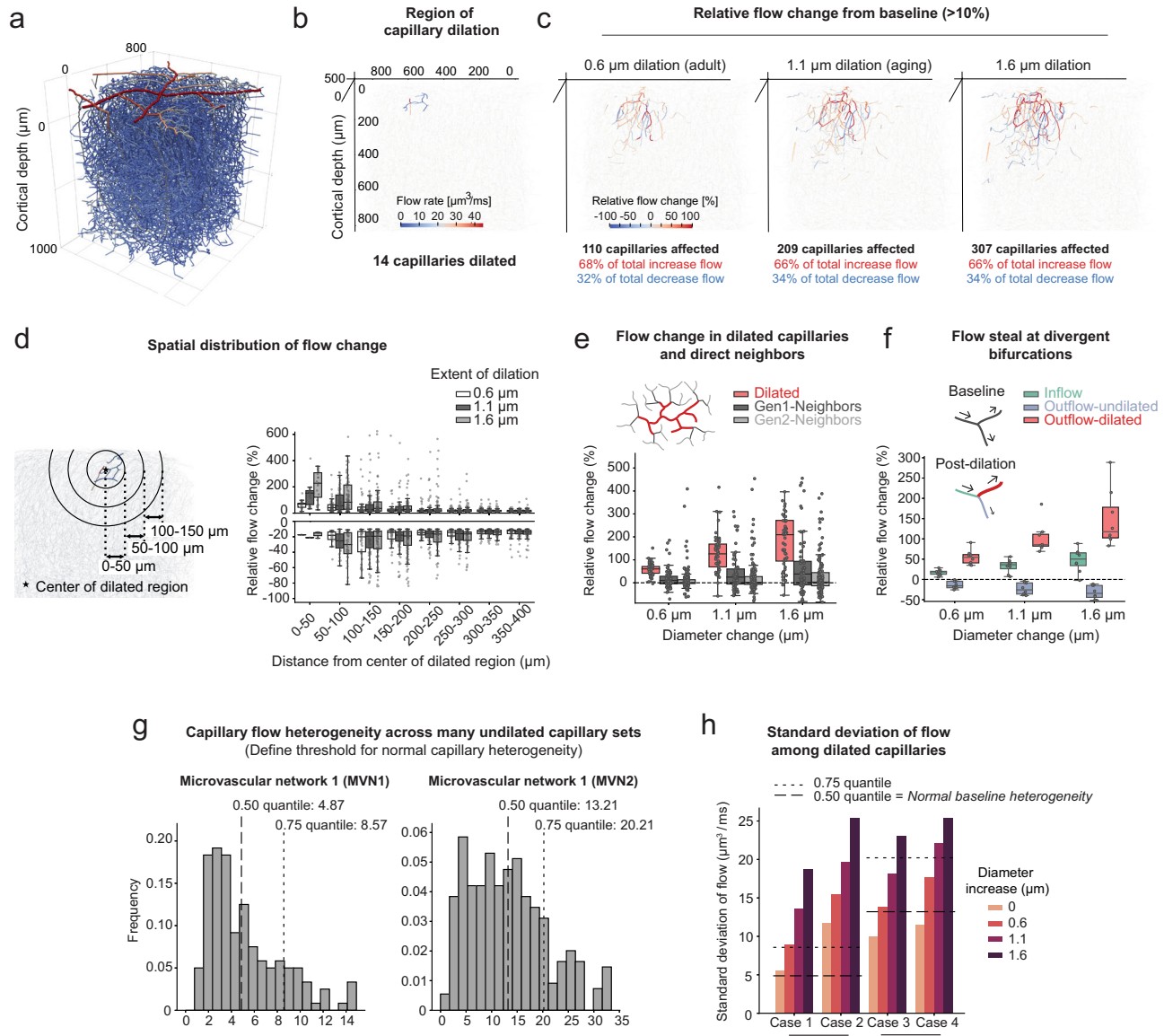

**Fig. 8 | Impact of local capillary dilations examined in silico. a** Microvascular network 1 (MVN1) from mouse parietal cortex. **b** Example of a typical region affected by pericyte ablation in MVN1. The color bar shows the flow rates in individual capillaries pre-ablation. **c** Relative flow changes >10% of baseline in the entire MVN1 for an increasing extent of dilation. Vessels with a relative flow change between −10% and +10% are not colored and depicted by the light gray lines. **d** Relative flow change >10% of baseline at different distances to the center of the dilated capillaries. $n(0.6) = 808$ vessels, $n(1.1) = 1114$ vessels, $n(1.6) = 1367$ vessels. **e** Relative flow changes in the dilated vessels and their direct neighbors (Gen1-Neighbors and Gen2-Neighbors, see schematic). $n(0.6, 1.1, 1.6) = 267$ vessels each. **f** Flow steal at divergent bifurcations with one dilated outflow

capillary. $n(0.6, 1.1, 1.6) = 24$ vessels each. Plots **d**–**f** show the relative flow change of individual vessels. The data of all four cases have been combined. For box plots, center = median, box bounds = upper (Q3) and lower (Q1) quartiles, whiskers = last data point within $Q1 − 1.5*(Q3 − Q1)$ and $Q3 + 1.5*(Q3 − Q1)$. **g** Distribution of standard deviations for numerous undilated capillary sets in upper cortex. The expected baseline heterogeneity of capillaries is defined as the median of all standard deviations. Left: MVN1 ($n$(sets)= 170, average set size = 14.5 ± 1.6), Right: MVN2 ($n$(sets)= 346, average set size = 14.6 ± 1.8). **h** Standard deviation of flow in the dilated capillaries at baseline and with increasing extent of dilation. Dashed lines show expected baseline heterogeneities at 0.50 and 0.75 quantile (see **g**).

were first collected for navigational purposes using a 4-X (0.16 NA) objective (Olympus; UPlanSAPO). We then switched to a 20-X (1.0 NA) water-immersion objective (Olympus; XLUMPLFLN) and used 975 nm for imaging of mural cells (tdTomato) and vasculature (FITC-dextran). For three channel imaging of tdTomato, GFP and Alexa 680-dextran we used 920 nm excitation. All imaging with the water-immersion lens was done with room temperature distilled water.

In the majority of studies, imaging was performed while the mice were anesthetized with 0.75-1.5% (MAC) isoflurane delivered in medical grade air through a custom nose cone. In a subset of studies, we imaged mice in the awake state, while head-fixed and free to run on a

low-resistance treadmill equipped with a motion sensor. In these studies that compared capillary diameters between anesthetized and awake state, the mice were first imaged under isoflurane, and then isoflurane was stopped, and the mouse was allowed to breathed medical air for a minimum of 15 min. Awake status of the mouse was evident by movement on the treadmill, and imaging data was captured during periods of rest between movements.

**Pericyte ablation**

Pericyte ablations were conducted with focused two-photon line-scan to specifically target individual pericyte somata[10,63]. This approach is

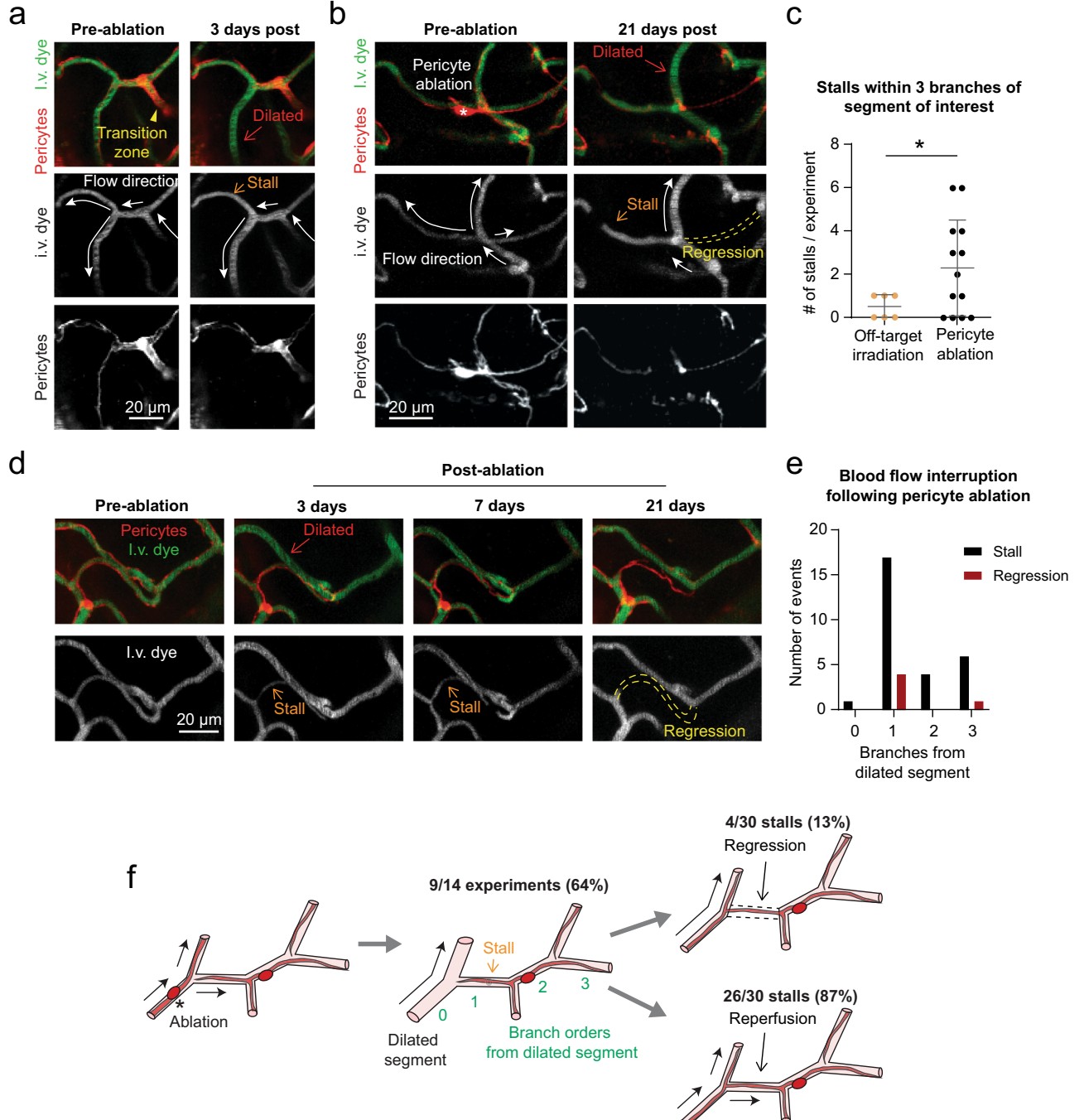

**Fig. 9 | Blood flow stalls and regression in capillaries neighboring dilation. a** A divergent bifurcation close to the arteriole-capillary transition zone. One branch of the bifurcation is dilated from pericyte loss 3 days post-ablation, and the resulting re-direction of flow creates a stall in the alternate, covered branch. Images are representative examples from 14 ablation experiments across adult and aged mice. I.v. dye = intravenous dye. **b** Example of blood flow defects at a capillary junction near the arteriole-capillary transition zone. At 21 days post-ablation, a central capillary is uncovered, dilated, and flowing. The other branches have stalled in flow or regressed. **c** Occurrence of stalls in triple off-target sham irradiation and triple pericyte ablation experiments. Unpaired *t* test with Welch's correction for unequal variances (two-sided), *t*(16.10) = 2.698; overall effect \**p* = 0.0158, for *n* = 6 off-target irradiation experiments (3 in adult mice, 3 in aged mice), *n* = 14 pericyte ablation experiments (7 in adult mice, 7 in aged mice). Data are shown as mean ± SD. **d** Additional example of a capillary regression occurring after a prolonged stall. **e** Plot of the relative location of stall and regression events following pericyte ablation, by branch order from a dilated segment. Observations from *n* = 14 ablation experiments (7 in adult mice, 7 in aged mice). **f** Schematic summarizing findings of blood flow interruption following pericyte coverage loss.

effective only for pericyte ablation in the upper ~200 μm of cerebral cortex. The ablative scan was performed at 725 nm. Ablative line-scans were applied in 20–30 s increments, for a total of 60–90 s per soma. The power ranged from 30 to 50 mW and was adjusted on a case-by-case basis taking into account the depth of the soma from the pial surface. To limit power input required for successful pericyte ablation,

all ablations were done within the first 100 μm of cortical depth beneath the pial surface (adult mice, 32.76 ± 15.66 μm; aged mice, 40.33 ± 16.51 μm from the pial surface. *p* = 0.1352 by unpaired *t* test for *n* = 21 pericytes from 6 adult mice, 21 pericytes from 6 aged mice). A successful ablation was confirmed on subsequent imaging days, if the targeted pericyte did not regain fluorescence. This indicated that cell

death was more likely than temporary photobleaching. Fluorescence recovery after photobleaching can occur within seconds to minutes after light irradiation, but loss of pericyte fluorescence caused by our ablations was permanent. Even with successful ablation, a dim imprint of the soma may remain for several days, as the dead cell is slowly cleared away. In unsuccessful ablations, the pericyte re-gains fluorescence over time in its soma and processes, as protein synthesis machinery replenishes cytosolic tdTomato. These cases, as well as cases where the ablation caused an immediate breach in the capillary wall, were excluded from the study.

For triple pericyte ablations, the day of ablation was considered "Day 0," and took place a minimum of 3 weeks after implantation of cranial window. Images of the area of interest were captured "pre-ablation" as well as "5 min post-ablation" on Day 0. The ablation regions were revisited and imaged at post-ablation days 3, 7, 14, and 21. At each imaging time point, a 70 kDa FITC-dextran i.v. dye was injected to simultaneously label the vasculature and test BBB permeability.

### Mural cells of different microvascular zones

These studies relied on mural cell morphology and vascular architecture in mouse cerebral cortex to define microvascular zones, as previously described[6,20,21]. We considered penetrating arterioles as branch order zero, and the 1st to 3rd branch order of penetrating arteriole offshoots as the "arteriole-capillary transition zone", which we previously called the "pre-capillary arteriole". We expected a greater likelihood of ensheathing pericytes in the transition zone, which are known to express high levels of α-SMA, and have protruding ovoid cell bodies and circumferential processes that completely cover the majority of the vessel wall. At 4th branch order and beyond, referred to as the "capillary zone", we expected occasional mesh pericytes and an abundance of thin-strand pericytes. This zone is α-SMA low/negative and thin-strand pericytes have fine processes that trail longitudinally on the endothelium. Since the number of branch orders within the arteriole-capillary transition zone varies slightly across penetrating arteriole offshoots[21], it was possible that mesh pericytes, when present, fell into either "transitional" and "capillary" groups. To preserving objectivity and consistency, we relied on both vascular branch order and mural cell morphology to categorize zones. On the venule side, we considered the ascending venule itself as the "venule zone," or "V0," which are covered by stellate-shaped venule SMCs.

### 10 kDa dye extravasation imaging and analysis

Aged PDGFRβ-tdTomato mice (35–40 g in weight) were implanted with chronic, skull-removed cranial windows. Three weeks after window implantation, double or triple pericyte ablations and sham irradiations were performed. Three days later, a single 30 μL bolus of 10 kDa FITC-dextran (5% w/v in sterile PBS; Sigma, FD10S) was injected retro-orbitally. Immediately following injection, image stacks were collected continuously from the affected area with an interval of 2 min over a period of 18 min. Fluorescence intensity over time was measured from ROIs using ImageJ/Fiji. Rates of parenchymal fluorescence intensity change were calculated using linear regression fitting over the first 10 min of data in MATLAB software.

### Microglial imaging and analysis

We bred PDGFRβ-tdTomato;CX3CR1-eGFP[+/-] mice to co-label mural cells in red and microglia in green. Mice were implanted with thinned skull windows and imaged 1–2 days post implantation[62]. During imaging, Alexa 680-dextran (2 MDa, custom conjugated)[64] was injected retro-orbitally to label the vasculature and imaging was performed at 920 nm excitation for three channel imaging. We examined the reaction of microglia to both the on-target pericyte ablations and off-target sham irradiations in adult and aged mice. Ablations and irradiations were performed one day after surgery (Day 0). The volume was imaged before ablation/sham, and again 30 min and 3 days after. Microglial reactivity was quantified as GFP intensity at the site of pericyte soma ablation or irradiation (focal ROI), as well as along the capillary territory occupied by the pericyte (territory ROI).

### Claudin-eGFP imaging and analysis

We bred PDGFRβ-tdTomato;Claudin5-eGFP mice to co-label mural cells in red and endothelial tight junctions in green. Mice were implanted with chronic, skull-removed cranial windows. During imaging, Alexa 680-dextran was injected retro-orbitally to label the vasculature and imaging was performed at 920 nm excitation. Images were captured prior to, and 5 min, 3 days, 7 days, and 14 days following on-target pericyte ablations or sham irradiations. The control group consisted of two sham irradiations and one failed pericyte ablation. Tight junction disassembly was analyzed by calculating the percentage of eGFP+ strands that contained gaps, and tight junction remodeling was analyzed by calculating the percentage of eGFP+ strands that contained protrusions or varicosities, following prior studies[27]. All data were normalized to pre-ablation. Only the tight junction strands on the vessel segment adjacent to the irradiation site were analyzed. Quantification was performed using image stacks in ImageJ/Fiji.

### Astrocyte endfeet imaging and analysis

Aged PDGFRβ-tdTomato mice received intracortical adeno-associated virus (AAV) injections during chronic, skull removed window implantation. A Nanoject III (Drummond) was used to inject 50 nL of pAAV.GfaABC1D.PI.Lck-GFP.SV40 (105598-AAV5; Addgene)[65] at -100 μm below the cortical surface. Mice were allowed to incubate with the virus for 3-weeks before imaging. Alexa 680-dextran was injected retro-orbitally and imaging was performed at 920 nm excitation. The apposition of astrocyte endfeet to the capillary wall was examined before and 3 days after pericyte ablation. Analysis of fluorescence intensity profiles was performed using ImageJ/Fiji.

### Quantification of 3-D vascular architecture

Totally capillary lengths and branchpoints were measured from 350 μm ($x$), 350 μm ($y$), and 90 μm ($z$; -10-100 μm from the pial surface) image stacks collected from adult and aged mice using Imaris x64 ver. 7.7.2 software.

### Quantification of pericyte remodeling

Pericyte process length over time was measured in 3-D using the Simple Neurite Tracer plugin for ImageJ/Fiji. To measure baseline process length, the full span of processes from soma to terminus was measured. All growth over time was plotted as a change in process length from baseline. Maximum growth rate was the greatest change in process length, in micrometers per day, between two imaging days. Average growth rate was calculated as the average of all post-ablation imaging timepoints. For average growth rate, only rates before pericyte-pericyte contact were included in the calculation. Pericyte-pericyte contact indicated completion of pericyte process remodeling. This was defined as the terminus of a process being <10 μm from another pericyte process.

### Quantification of capillary diameter

Capillary segment diameters were measured with the ImageJ/Fiji macro Vasometrics, which provides unbiased diameter measurements across a vessel segment of interest (full-width at half-max fluorescence intensity) and reports an average diameter[66].

### Acquisition and quantification of capillary flow data

Line-scans were performed with 2MDa FITC-dextran, at a wavelength of 800 nm. Line-scans were obtained for each capillary of interest with a single line transecting the center of the vessel. We calculated RBC flux by manually counting the number of blood cell streaks (over ~1.3 s of

scan time) and normalizing all numbers to represent flux in a period of 1 s. The angle of the streaks in relation to the direction of the scan was used to determine RBC flow direction. Line-scans were presented to an independent rater in a blinded fashion (S.S.), to reduce bias.

## Statistics for in vivo data

All statistical analyses were performed with GraphPad Prism version 8 software or MATLAB (R2018a). Details of specific tests can be found in the Figure legends. For all unpaired $t$-tests, $F$ tests for equality of variance were performed. When statistically significant $F$ tests were observed, unpaired $t$-test with Welch's correction for unequal variance was used.

## Microvascular networks for in silico modeling

The vectorized microvascular networks used for blood flow modeling were acquired in a previous study by ref. [67]. In these data, each vessel is represented by an edge with a given length and diameter. The edges are connected at bifurcation (graph vertices), such that a fully connected microvascular network is obtained.

Both microvascular networks used for in silico modeling in this work were acquired from the vibrissa primary sensory (vS1) cortex of C57/BL6 male mice. Microvascular network 1 (MVN1) and 2 (MVN2) were within a tissue volume of ~1.6 mm³ and ~2.2 mm³ and contained ~12,100 and ~19,300 vessels, respectively. The vessels are labeled as pial arteries, descending arterioles, capillaries, ascending venules, and pial veins; 96% (MVN1) and 94% (MVN2) of all vessels were capillaries.

## Blood flow modeling with discrete RBC tracking

Analysis of in silico data was performed in Python (version 2.7). The numerical model to simulate blood flow with discrete RBC tracking in realistic microvascular networks was previously described[35]. We briefly summarize the key aspects of the modeling approach here, but more detailed descriptions are available in prior studies[35,68]. The modeling approach is based on the small Reynolds number (Re < 1.0 for all vessels) in the microvasculature. As such the flow is laminar and mostly even in the Stokes regime, which allows description of the flow in individual vessels by Poiseuille's law. The flow rate $q_{ij}$ in vessel ij between node i and j is computed by:

$$q_{ij} = \frac{p_i - p_j}{R_{ij}^e} = \frac{\pi D_{ij}^4}{128\, L_{ij}\, \mu\, \mu_{rel}^e} (p_i - p_j)$$

where $D_{ij}$ and $L_{ij}$ are the vessel diameter and the length and $p_i$ and $p_j$ are the pressure at node i and j, respectively. $\mu$ is the dynamic plasma viscosity and $\mu_{rel}^e$ the relative effective viscosity, which accounts for the presence of RBCs and is computed as a function of local hematocrit and vessel diameter as described in ref. [69].

To compute the local hematocrit we track individual RBCs through the MVNs. Hereby, we need to account for the Fahraeus effect (RBC velocity is larger than bulk flow velocity) and the phase separation (RBCs partition with a different ration than the bulk flow at divergent bifurcations)[69]. The phase separation effect is also called Zweifach-Fung effect[33] and describes the phenomenon that at divergent bifurcations the daughter vessel with the larger flow rate receives a larger fraction of RBCs. At divergent bifurcations with a diameter >10 μm the phase separation is described by empirical equations[69]. At smaller diameters RBCs propagate in single file flow and consequently the RBC motion can be approximated by assuming that the RBCs follow the path of the largest pressure force[33,35]. It is important to note that the RBC distribution and therewith the flow field fluctuate in time and that the RBC distribution impacts the local flow field[68,70]. In this study, we use the time-averaged flow field to compare changes in response to locally altered capillary diameters. The time-averaged flow field is computed by averaging over 15.4 s. The RBC flux per vessel is computed from the product of the time-averaged flow rate and discharge hematocrit, where the discharge hematocrit is a function of the local tube hematocrit and the vessel diameter[69]. The pressure boundary conditions and the inflow hematocrit of 0.3 are kept constant for all simulations. Further details on assigning suitable pressure boundary conditions are available in ref. [35].

## Analyses of in silico data

The capillaries chosen for dilation were selected to mimic the in vivo scenario as closely as possible. They were chosen in the upper 100 μm of cortex and were beyond four branch orders from the main branch of the penetrating arteriole (0th order), and beyond one branch from the ascending venule main branch. In vivo, between 12 and 15 capillaries were affected by ablating three contiguous pericytes, which corresponds to an uncovered capillary length of 688 ± 70 μm. The pre-ablation characteristics of these capillaries as measured in vivo were: (1) diameter: 3.04 ± 0.59 μm (range: 2.17–4.81 μm), (2) RBC flux: 114 ± 73 RBCs/s (range: 17–381 RBCs/s) and (3) RBC velocity: 1.11 ± 0.63 mm/s (range: 0.19–2.70 mm/s).

We used these pre-ablation characteristics to choose representative "base capillaries" around which dilations seen after in vivo pericyte ablation were mimicked. To avoid being too close to the cortical surface in silico the base capillary had to be at a cortical depth between 20 and 120 μm. In addition to these selection criteria, we added that the base capillary needed to be at least two branches apart from the boundary of the simulation domain and that the Euclidean distance of the base capillary be smaller than 0.9 times the mean Euclidean distance of all vessels to the center of the MVN in the x-y-plane. The latter two constraints ensured that the simulation results were not affected by the pressure boundary conditions at the domain boundary. The selection criteria for the base capillary are summarized in Supplementary Table 1. We found that 193 and 382 capillaries fulfilled the selection criteria in MVN1 and MVN2, respectively. From each subset we randomly chose 7 base capillaries.

Subsequently, 12–15 capillaries around the base capillary need to be selected to mimic the extent of dilations observed in response to the ablation of three pericyte somata. To attain this goal, we first added the capillaries directly adjacent to the upstream bifurcation of the base capillary (generation 1 neighbors) to the set of affected capillaries. Subsequently, the capillaries connected to the generation 1 neighbors were added. The procedure was continued until >= 6 capillaries had been added to the set of affected capillaries. This sequential approach was also initiated at the downstream node of the base capillary until at least 13 capillaries were selected in total. Vessels that were: (1) labeled as main branch (0th order), (2) less than two branches apart from a descending arteriole main branch and (3) less than two branches apart from the domain boundary could not be added to the set of affected capillaries. Thus, the total number of affected capillaries and the number of capillaries added on the upstream and downstream side of the base capillary varied for different base capillaries. Supplementary Table 2 summarizes the baseline characteristics for 7 distinct base capillaries in MVN1 and MVN2. Two cases per MVN were chosen for further analysis (referred to as Cases 1–4 in the Figures).

To mimic the alterations in response to pericyte ablation in vivo, the set of affected capillaries was dilated. Three scenarios were tested: (1) Dilation by 0.6 μm (average response to pericyte ablation in adult mice), (2) Dilation by 1.1 μm (average response in aged mice), and (3) Dilation by 1.6 μm (extreme scenario in aged mice). For all four cases and for each dilation scenario the blood flow simulation was repeated and the time-averaged flow field computed.

## Comparing pre- and post-ablation characteristics in silico

In the analyses of the simulation results, we focused on relative flow changes in response to dilation of the capillaries affected by pericyte ablation. As previously stated, due to the presence of RBCs, the flow field fluctuates in time[70]. These fluctuations can also affect the time-

averaged flow field. To compute representative relative changes and to avoid large relative changes in capillaries with low baseline flow rate, we employ an absolute threshold of $0.1\,\mu m^3/ms$. Changes below this threshold were set to 0. These vessels represented a minority, as 2.6% (MVN1) and 0.7% (MVN2) of all vessels had a time-averaged flow rate $<0.1\,\mu m^3/ms$. More details on the definition of a suitable threshold are available in ref. 71.

To analyze the impact of the local capillary dilations across larger regions of the MVN, we computed the relative flow changes in analysis spheres around the center of the dilated region (Fig. 8d). The center of the dilated region was defined as the averaged coordinate of all bifurcations of the dilated capillaries. Subsequently, we computed the Euclidean distance of each vessel to the center of the dilated capillaries and grouped the vessels based on the resulting distance to center. Additionally, we separated vessels with flow increase and decrease. Only relative changes >10% from baseline were considered for this analysis.

To more closely inspect relative changes in the vicinity of the dilated vessels we compare the relative flow change in: (1) the dilated capillaries, (2) the capillaries adjacent to the dilated capillaries (Gen1-Neighbors), and (3) the capillaries adjacent to the Gen1-Neighbors (Gen2-Neighbors)(Fig. 8e). We further examined flow changes at divergent bifurcations with one dilated and one un-dilated outflow vessels (Fig. 8f). Across the four cases, six such configurations could be identified.

To compare flow heterogeneity in the set of affected capillaries to the expected baseline heterogeneity, we calculate the standard deviation of flow for 193 and 382 different sets of capillaries for MVN1 and MVN2, respectively. These sets of capillaries contained 12–15 capillaries each and were defined similarly as the set of affected capillaries to mimic pericyte ablation. The distribution of standard deviations for MVN1 and MVN2 are shown in Fig. 8g. The median standard deviation of such a set of capillaries is used as reference for the expected baseline heterogeneity.

To examine for changes in the number of stalled or no-flow capillaries after focal dilations, we set a threshold value at the lowest 5% of all time-averaged flow rates in the upper $200\,\mu m$ at baseline. For MVN1 and MVN2 this corresponds to a flow rate of $0.37\,\mu m^3/ms$ and $0.92\,\mu m^3/ms$, respectively. For all scenarios (baseline, diameter change = $0.6\,\mu m$, $1.1\,\mu m$, and $1.6\,\mu m$) we count the number of stalled/low flow capillaries (i.e., with a time-averaged flow rate smaller than the lowest 5% threshold) in the dilated, generation 1 (Gen-1) and generation 2 (Gen-2) neighbors (Supplementary Fig. 12c). In a similar manner, we quantify the number of low flow stalled capillaries within $200\,\mu m$ distance from the center of focal dilation (Supplementary Fig. 12d).

### Reporting summary

Further information on research design is available in the Nature Research Reporting Summary linked to this article.

## Data availability

Source data are provided with this paper. Raw image files are stored on servers at Seattle Children's Research Institute owing to their large size. These raw data can be provided from the corresponding author upon request.

## Code availability

The in silico data generated for this study together with analyses scripts and instructions have been deposited on Zenodo: https://doi.org/10.5281/zenodo.7038939.[72] Additional details regarding the simulation code used to produce the in silico results are available in the original publication of the in silico model[35]. Further details and explanations are available from F.S. upon request.

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

## Acknowledgements

Our work is supported by grants to A.Y.S. from the NIH/NINDS (NS106138, NS097775) and NIH/NIA (AG063031, AG062738), the American Heart Association (14GRNT20480366), Alzheimer's Association NIRG award (2016-NIRG-397149), a scholarship for Research in the Biology of Aging to A.A.B. from the American Federation for Aging Research, a post-doctoral fellowship from the American Heart Association (20POST35160001) to V.C.S., and by a grant from the Swiss National Science Foundation (310030_182703) to B.W. We appreciate the helpful comments and discussion of B. Gumbiner, C. Cowan, N. Bhat, L. Faulhaber, and J. Gust. We thank D. Agalliu for generously providing Claudin5-eGFP mice.

## Author contributions

Conceptualization and experimental design by A.A.B. and A.Y.S. Experiments were conducted by A.A.B., S.S., V.C.S., C.N., and A.Y.S. Data analysis and statistics were performed by A.A.B., S.S., V.S.C., C.N., and A.Y.S. In silico studies and analyses were performed by FS and BW. The manuscript was written by A.A.B., F.S., and A.Y.S. with contributions from S.S., V.C.S., C.N., B.W., and M.W.M.

## Competing interests

The authors declare no competing interests.
