## [Peer Review File · Nature Communications]

REVIEWER COMMENTS

Reviewer #1 (Remarks to the Author):

“Deficiency in pericyte remodeling as a basis for impaired capillary flow and structure during brain aging”

By Berthiaume et al

Pericytes are cells that grow along and enwrap capillaries, the smallest vessels in the brain, and play important roles in maintenance of vascular integrity and modulation of blood flow. Pericyte loss or dysfunction is implicated in multiple neurological diseases including Alzheimer’s disease. Previously the authors have shown using longitudinal in vivo two-photon imaging that ablation of single pericytes in vivo leads to vessel dilation in the territory covered by the lost pericyte. And, that neighboring pericytes extend their processes to re-cover the ablated pericyte’s territory, restoring vascular tone. Here the authors extend this work to probe the plasticity characteristics of pericytes and underlying vasculature in adult (3-6 mo old) and aged (18-24 mo old) mice by ablation of sets of three pericytes in a localized area. Similar to their previous work, the authors found that neighboring pericytes re-covered most of the territories of the lost pericytes rapidly within three weeks in adult mice. However, re-coverage proceeded more slowly and was incomplete in aged mice, even at extended time points. Interestingly, the authors reported observing transient “microbends” in vessels as new pericyte processes grew along them. The authors hypothesized that the new pericyte processes applied tension to the vessels as they grew along them creating the bends.

Loss of sets of pericytes led to dilation of the uncovered vessels, and were accompanied by aberrant blood flow changes in the uncovered vessel territories and surrounding upstream and downstream vessels. Diameter changes were significantly larger in the aged mice compared to adult mice, and the vessels in aged mice tended to remain slightly dilated after restoration of pericyte coverage. Blood flow tended to be faster in the uncovered vessels, but slower in the surrounding pericyte covered vessels. Similar results were confirmed using in silico vascular models. In both adult and aged mice, the differences in blood flow related to the ablated pericyte territories led in some cases of stalled vessels and occasional permanent loss of vessel segments.

Finally, the authors investigated the effect of ablation of different types of pericytes (e.g. mesh pericytes near arterioles compared to thin strand pericytes on higher order capillaries). Here the authors found that neighboring pericytes extended processes into ablated pericyte territories in patterns that matched that of the original ablated pericyte, regardless of the neighboring pericyte’s type.

This is an interesting study that builds on this group’s previous work by investigating the effect of aging on pericyte plasticity and regulation of vascular tone. The information presented may have implications for understanding how microvascular dysfunction and loss occurs in healthy aging and diseases. The real-world example of vascular zonation exhibited by pericyte processes growing across differing vascular zones is unique and elegant. The manuscript would be improved by addressing the following:

Major:

1. The authors note that the number of pericyte soma were similar between the adult and aged mice. What is the pericyte vascular coverage for the adult and aged mouse groups? Could the aged mice have started out with lower pericyte coverage, since pericyte loss can occur with aging? This could have implications for interpretation of the results.
2. The authors show that pericytes in aged mice are less likely to extend processes along vessels into the territories of ablated pericytes than younger adult mice. Could this be due to change in expression of Pdgf-BB and/or Pdgfrb by endothelial cells and pericytes, respectively, with age? This ligand and receptor are important for maintaining pericytes at the vessel wall, and would speak to

a potential mechanism behind the observed changes with age.

3. The authors focus on the superficial layer of the brain parenchyma (top ~100um). The authors should discuss the potential for translation of these observations to deeper cortical layers, as the vascular density varies with cortical depth. This also has implications for tissue oxygenation, which the authors touch on briefly in the discussion (ref. 28), but should also note that adult animals with pericyte loss exhibit deficits in tissue oxygenation and hypoxic pockets, particularly in deeper cortical layers (Nat Neurosci. 2017 Mar;20(3):406-416) and tissue hypoxia after substantial acute pericyte loss (Nat Neurosci. 2019 Jul;22(7):1089-1098).

4. The authors show that vessel branch order/location determines the format of pericyte process coverage when pericytes re-cover territories of ablated pericytes (e.g. zonation). This is an important observation that is somewhat lost in the noise of the manuscript as presented. It has been an open question since the Betsholtz lab observed in their single cell RNA sequencing study (Nature. 2018 Feb 22;554(7693):475-480) that describes multiple endothelial cell profiles that correspond to different vascular zones but only one pericyte genetic profile. Yet, morphologically distinct pericyte populations have been observed in multiple studies, including the author's works. The authors should consider adding a discussion of their data in regards to these RNA sequencing results.

5. The authors conclude that there are no microvascular leakages detected after triple pericyte ablation, based on in vivo imaging of 70kDa fluorescent dextran, and lack of microglial activation. This conclusion is perhaps premature, as 70kDa dextran is rather large and would likely only detect leakage due to significant breakdown of the BBB. A smaller sized tracer should be used to confirm these observations. Furthermore, lack of microglial activation may not reflect lack of vascular leakage. For example, one study did not observe significant microglial activation after acute brain-wide pericyte ablation, even though loss of pericytes resulted in increased BBB permeability and extravascular IgG deposits (Nat Neurosci. 2019 Jul;22(7):1089-1098).

6. Please clarify what is meant by "transient" capillary stalls. For example, transient stalls were those detected only for a few minutes in a recent study (ref. 25). From figure 8d, it seems like the particular vessel noted experienced a series of stalling events, given the different locations of the RBC shadows in the vessel over different days.

Minor:

7. Please clarify the sentence on lines 117-118: "...inducing endothelial contact with multiple pericytes."

8. Figure 6a: The purple color is hard to see overlaid on the vessels.

9. Please provide a reference for the statement about the observation of string vessels in aging, Alzheimer's disease and VCID on lines 398-399.

10. Consider rephrasing the statement on lines 428-430 or providing additional citations to support it. Aberrant contraction of pericytes is a mechanism that can lead to blood flow impairment, but whether it is the "primary" mechanism is unclear.

Reviewer #2 (Remarks to the Author):

The aetiology of age-and disease related vascular changes is an intriguing subject as these changes are likely central to the pathogenesis of a number of neurodegenerative disorders. Based on previous work by the same group, this study investigates the consequences of selective cortical pericyte loss in adult and aged mice, whereby the novelty is the comparison to the aged brain.

Using a photo-thermal pericyte ablation, the authors perform a triple pericyte ablation in mice and

examine the subsequent remodelling capacity of pericytes and its consequences for blood-brain barrier leakage, vascular tone and capillary flow comparing adult and aged mice. The study is well-performed, adequate methods are used and the manuscript is very well written.

The authors document a delayed and incomplete pericyte remodelling in aged mice resulting in reduced capillary tone. In both groups, pericyte loss leads to capillary flow abnormalities.

Major comments

1. Alterations in brain capillary functions are a well known phenomenon in the aging brain, however, the authors find no difference in cortical capillary structure in terms of capillary length, number of capillary junctions, number of pericyte somata (Figure 1 c, d and S1e) and in vascular tone between adult and aged mice.

-Was there any difference in pericyte (PDGFR β -tdTomato) density?

-Were there age-related vascular changes in other brain regions, as this has been reported by other groups?

2. It seems that the ability to extend processes varies highly (0-200 μ m) between individual pericytes. In aged mice, more capillary surface remains uncovered at 21 days consistent with a lower number of pericyte-pericyte contacts. The authors conclude that the recruitment mechanism of pericytes is impaired in aged mice.

However, the data are incomplete to support this statement:

Despite the above statement, the authors show that there is no significance difference between groups for process length at day 21 (Figure 3 h), even though the processes have a significantly slower growth rate.

So why is there a larger capillary area that is still uncovered in aged mice? Was there a larger area uncovered after the lesion in the first place? Figure 3 I suggest that already at time point 0, both groups had very different starting positions in terms of uncovered vessel area. Could this be due to the choice of pericytes to be ablated: The authors state that the pericyte process length not only greatly varied between pericytes, but also determined their growth capacity. In figure S3c they show that there was no difference in baseline length of pericyte processes between groups. However, figure S3c does not show the baseline length of the lesioned pericytes. If only 3 pericytes are depleted per animal, the choice is of course important if it determines their growth capacity. This data should be possible to retrieve from pre-lesion images and should be presented to control for any selection bias. Such differences in pericyte selection would also account for the larger uncovered area in aged mice at the starting point.

The second determinant of pericyte growth is splitting which is dependent on the number of bifurcations in the depleted territory. The authors state that there was no difference in the number of bifurcations, but data supporting this are not presented.

3. The authors state that they found no evidence of pericyte proliferation or cell recruitment in their paradigm: How was this assessed since the imaging interval is days to weeks?

Did the authors stain for cell division markers? How was pericyte recruitment excluded?

4. Differences across microvascular zones

Here the authors compare 3 different anatomical regions in 2 different groups. Judging from Figure 4 c, aged mice extend longer processes along venules than adult mice. However, the graphs are separated by groups, whereas this was not done in Figure 4 f.

All conditions should be presented together and analysed by 2-way ANOVA.

5. Blood Brain Barrier (BBB) leakage is an important consequence of vascular alterations and seen as one of the implications of pericyte loss and a large contributor to neurodegeneration. In the current study paradigm, the authors inject only one dye of a large molecular weight and state that the pericyte loss was insufficient to cause BBB leakage. However, BBB leakage can be subtle, especially if the pericyte loss is so focussed. It would be extremely interesting to see whether any subtle BBB leakage can be reversed upon pericyte remodelling. I suggest the authors inject a smaller molecule and report the circulation time before imaging. Circulation time, details of the volume, concentration, supplier and average weight of the mice need to be reported for the dyes injected.

Minor comments: Please correct sentences

Line 61: perfusion blood cells

Line 83: on how pericyte loss related in deficiencies

1. Thermal photo method for pericyte ablation. The researchers assumed that the focused photon on the pericyte soma causes the ablation of the pericyte. They exclude the possibility of photobleaching rather than killing pericyte cells, based on that the loss of pericyte signal was permanent while photobleaching is temporary. What is the time window of the temporary bleaching? Please support with references.

2. Quantification of the vascular length

The authors used the ImageJ (AngioTool) to estimate vessel length. How was the 90 um thick image processed? Is the analysis done in 3D or 2D? If 2D, how do they deal with possible overlapping between the vessels at different depths when converted to max projection.

Reviewer #3 (Remarks to the Author):

The manuscript presents a study of correlation between deficiency in pericyte remodeling and alterations in microvascular flow in an aging brain. Pericyte recovery is found to slow down with age in a mouse model. Furthermore, pericyte loss appears to result in local vasodilation of capillaries, which affects blood flow distribution. As a result, alterations in blood flow cause remodeling of microvascular structure, potentially leaving certain portions of the brain tissue underperfused. In addition, some capillaries may receive a little amount of erythrocytes due to blood flow alterations.

I think this is a very interesting hypothesis, which clearly deserves an investigation. The presented data support that there is a correlation between pericyte loss and the remodeling of microvasculature, however, it is not clear whether it would be the main cause or a contribution to the vascular remodeling. Therefore, I would be a bit careful with some of the statements. Below I provide a few suggestion which I hope will be useful for the authors to improve the manuscript.

1) The authors often use the word "maldistribution" referring to altered blood flow. I agree that blood flow is changed, but it is much less clear whether blood flow becomes kind of malfunctional or pathological.

2) The blood steal effect is called the Zweifach-Fung effect, which states that at bifurcations, daughter branches with a larger flow rate generally receive a larger erythrocyte content. This needs to be properly explained and cited.

3) The presented data show that capillary regression is strongly associated with "capillary stalls" (no significant flow). However, no difference in stall or regression frequency was found between different age groups. Since your proposition of microvascular remodeling is mainly associated with "capillary stalls", then how would it occur differently for different ages?

4) Does brain microvasculature have also functional vasodilation/vasoconstriction, which generally takes place at the arteriolar (pre-capillary) part of microvasculature? How do you exclude such effects?

5) Would angiogenesis kick in if some portions of brain tissue are becoming hypoxic? Or would it be irrelevant for some reason?

6) I think the main difficulty in simulations is to assign proper boundary conditions for numerous inlets and outlets. How do you do this? Do you make any 'semiquantitative' comparison with experiments? How sensitive and robust your simulation results to the choices of boundary conditions made? Do you see any capillary stalls in simulations?

7) I think it would be interesting to look at an effective 'oxygen delivery' in simulations, which would be a product of local erythrocyte content and flow. For example, your model implements bifurcation laws (or the Zweifach-Fung effect), so that erythrocyte content can be locally assessed. Do you observe significant erythrocyte steals at some bifurcations? How strong such changes can be?

RESPONSE TO REVIEWER COMMENTS

We thank the reviewers for their thoughtful comments and careful reading of our manuscript. We have revised the manuscript to address all the comments to the best of our ability. In the process, we performed several new experiments and analyses, producing a more comprehensive and balanced manuscript. In particular, we provide new data confirming that there is no overt leakage of the BBB at the scale of pericyte loss we have induced. In addition, we include new analyses of *in silico* data showing that the basal heterogeneity of capillary flow in regions of pericyte loss is much higher than a normal range seen in cortex. This strengthens the concept that flow disturbances with pericyte loss could lead to abnormal effects blood flow through capillary networks.

We have added one main figure to focus on the topic of BBB function (**Figure 5**). We have also added new Supplementary figures for experiments looking at endothelial tight junction structure (**Supplementary Fig. 7**) and astrocytic endfeet (**Supplementary Fig. 8**) following pericyte loss *in vivo*. Substantive changes and additions to the manuscript are shown in blue text.

Reviewer #1 (Remarks to the Author):

“Deficiency in pericyte remodeling as a basis for impaired capillary flow and structure during brain aging” By Berthiaume et al

Pericytes are cells that grow along and enwrap capillaries, the smallest vessels in the brain, and play important roles in maintenance of vascular integrity and modulation of blood flow. Pericyte loss or dysfunction is implicated in multiple neurological diseases including Alzheimer’s disease. Previously the authors have shown using longitudinal *in vivo* two-photon imaging that ablation of single pericytes *in vivo* leads to vessel dilation in the territory covered by the lost pericyte. And, that neighboring pericytes extend their processes to re-cover the ablated pericyte’s territory, restoring vascular tone. Here the authors extend this work to probe the plasticity characteristics of pericytes and underlying vasculature in adult (3-6 mo old) and aged (18-24 mo old) mice by ablation of sets of three pericytes in a localized area. Similar to their previous work, the authors found that neighboring pericytes re-covered most of the territories of the lost pericytes rapidly within three weeks in adult mice. However, re-coverage proceeded more slowly and was incomplete in aged mice, even at extended time points. Interestingly, the authors reported observing transient “microbends” in vessels as new pericyte processes grew along them. The authors hypothesized that the new pericyte processes applied tension to the vessels as they grew along them creating the bends.

Loss of sets of pericytes led to dilation of the uncovered vessels, and were accompanied by aberrant blood flow changes in the uncovered vessel territories and surrounding upstream and downstream vessels. Diameter changes were significantly larger in the aged mice compared to adult mice, and the vessels in aged mice tended to remain slightly dilated after restoration of pericyte coverage. Blood flow tended to be faster in the uncovered vessels, but slower in the surrounding pericyte covered vessels. Similar results were confirmed using *in silico* vascular models. In both adult and aged mice, the differences in blood flow related to the ablated pericyte territories led in some cases of stalled vessels and occasional permanent loss of vessel segments.

Finally, the authors investigated the effect of ablation of different types of pericytes (e.g. mesh pericytes near arterioles compared to thin strand pericytes on higher order capillaries). Here the authors found that neighboring pericytes extended processes into ablated pericyte territories in patterns that matched that of the original ablated pericyte, regardless of the neighboring pericyte's type.

This is an interesting study that builds on this group's previous work by investigating the effect of aging on pericyte plasticity and regulation of vascular tone. The information presented may have implications for understanding how microvascular dysfunction and loss occurs in healthy aging and diseases. The real-world example of vascular zonation exhibited by pericyte processes growing across differing vascular zones is unique and elegant. The manuscript would be improved by addressing the following:

Major:

1. The authors note that the number of pericyte soma were similar between the adult and aged mice. What is the pericyte vascular coverage for the adult and aged mouse groups? Could the aged mice have started out with lower pericyte coverage, since pericyte loss can occur with aging? This could have implications for interpretation of the results.

We re-examined capillary structure, pericyte density, and pericyte coverage in 3D (previous analyses were of 2D projections). These data are updated in **Supplementary Fig. 1c-f**, and appended below for convenience. We continue to see no difference in capillary structure or pericyte metrics in regions we have imaged, which are restricted to upper layers of cortex.

Capillary rarefaction^{1,2} and pericyte loss³ reported in brains of aged mice is more prominent in deeper brain regions distant to the arteriole perfusion source, such as callosal white matter. As will be discussed further below (comment 3), we think this is the reason why we do not see age-related loss of capillary structure and pericytes in upper cortex.

2. The authors show that pericytes in aged mice are less likely to extend processes along vessels into the territories of ablated pericytes than younger adult mice. Could this be due to change in expression of Pdgf-BB and/or Pdgfrb by endothelial cells and pericytes, respectively, with age? This ligand and

receptor are important for maintaining pericytes at the vessel wall, and would speak to a potential mechanism behind the observed changes with age.

This is a great question. We are conducting a separate study on how pericyte-endothelial signaling contributes to pericyte remodeling (including PDGF-BB/PDGFR β signaling) and these studies will use pharmacology and pericyte-specific genetic approaches. This will hopefully lead to targets for promoting pericyte remodeling. However, it is a long-term project requiring characterization of new pericyte-specific lines, triple transgenics, and aging that extends beyond the scope of the current work.

For the revision, we address this excellent question by adding details to the Discussion. For example, a recent study showed that genetic ablation of PDGF-BB in endothelial cells during adulthood leads to age-related pericyte loss and BBB defects.⁴ This suggests that PDGFB signaling is still involved in capillary maintenance in adulthood and in the aged brain. However, there are other possible pathways involved that we plan to explore, including FGF, VEGF and Ang1-Tie2 signaling.

3. The authors focus on the superficial layer of the brain parenchyma (top ~100 μ m). The authors should discuss the potential for translation of these observations to deeper cortical layers, as the vascular density varies with cortical depth. This also has implications for tissue oxygenation, which the authors touch on briefly in the discussion (ref. 28), but should also note that adult animals with pericyte loss exhibit deficits in tissue oxygenation and hypoxic pockets, particularly in deeper cortical layers (Nat Neurosci. 2017 Mar;20(3):406-416) and tissue hypoxia after substantial acute pericyte loss (Nat Neurosci. 2019 Jul;22(7):1089-1098).

The optical ablation approach we have used is only effective in the upper layers of cortex (upper 200 μ m), and we have now stated this limitation in the revised Methods. The reviewer makes an excellent point that deeper tissues might experience different consequences with pericyte loss. Since capillary density and pericyte numbers seem to be higher in cortical layers 4 and 5⁵, we conjecture that these tissues might be more resilient to pericyte loss and exhibit more robust capillary remodeling.⁵ Conversely, capillary density is low in callosal white matter, and studies have shown that aging reduces capillary density in the white matter of mice.^{1, 6} Thus, white matter is likely most vulnerable during pericyte loss, as highlighted in prior studies.⁷

In unpublished studies using deep two-photon imaging of adult and aged mice, we have observed selective loss of capillary density and perfusion in cortical layer 6 and callosum with aging. This is logical because these vessels are more distant to the arterial perfusion source from the pia. We also expect that this density loss corresponds with reduced tissue oxygenation, as indicated by the reviewer. However, this remains to be determined in our ongoing studies.

4. The authors show that vessel branch order/location determines the format of pericyte process coverage when pericytes re-cover territories of ablated pericytes (e.g. zonation). This is an important observation that is somewhat lost in the noise of the manuscript as presented. It has been an open question since the Betsholtz lab observed in their single cell RNA sequencing study (Nature. 2018 Feb 22;554(7693):475-480) that describes multiple endothelial cell profiles that correspond to different vascular zones but only one pericyte genetic profile. Yet, morphologically distinct pericyte populations have been observed in multiple studies, including the author's works. The authors should consider adding a discussion of their data in regards to these RNA sequencing results.

Thank you for this important comment. In a recent review⁸, we specifically delved into this topic. We drew relationships between the transcriptomically-defined mural cell groups from the Betsholtz lab and mural cell morphologies/functions commonly described in *in vivo* imaging studies. In that review, we suggested that ensheathing pericytes are aaSMCs, mesh and thin-strand pericytes are PCs, and venular pericytes and SMCs are vSMCs, as categorized in Vandlandewijck et al.⁹ We suggest that the concept of pericytes having one general transcriptional profile may be misinterpreted as one functionally homogeneous cell type, when in fact a range of transcriptional states could reflect differences in functionality across microvascular zones.

For this manuscript, we exercise caution in drawing strong links between RNA transcriptomic and *in vivo* studies at this time. The cell categories from single cell transcriptomics are derived from whole brain vasculature, and our studies are restricted to superficial cortex. Differences in mural cell zonation across cortical depths and brain regions is still not well understood. Further, the transcriptionally-defined cell categories have not been thoroughly mapped onto the 3-D vascular architecture of cortex, which is central to *in vivo* imaging studies.

What we can say for certain from our study is that pericytes from different zones can all remodel their processes, but ensheathing and capillary pericytes exhibit reduced remodeling capacity with age. Also, pericyte processes can change their morphologies based on the vascular zones they contact. However, we do not know if these differences are intrinsic properties of the pericyte types, or a result of being exposed to zone-specific signals from the endothelial cells.

5. The authors conclude that there are no microvascular leakages detected after triple pericyte ablation, based on *in vivo* imaging of 70kDa fluorescent dextran, and lack of microglial activation. This conclusion is perhaps premature, as 70kDa dextran is rather large and would likely only detect leakage due to significant breakdown of the BBB. A smaller sized tracer should be used to confirm these observations. Furthermore, lack of microglial activation may not reflect lack of vascular leakage. For example, one study did not observe significant microglial activation after acute brain-wide pericyte ablation, even though loss of pericytes resulted in increased BBB permeability and extravascular IgG deposits (Nat Neurosci. 2019 Jul;22(7):1089-1098).

We agree that our data with 70 kDa FITC-dextran does not rule out potential leakage of smaller molecules. In our past work, we had used very small dyes (~1 kDa, cadaverine) and reported no leakage 3 days after ablation of single pericytes, following the approach of Lou and Nedergaard¹⁰ and Armulik et al.¹¹ However, the limitation was that the dye stayed in circulation for a matter of seconds making it easy to miss slow and subtle leakage.

In response to the reviewer's important comment, we performed additional studies in aged mice to examine whether 10 kDa FITC-dextran leakage occurs at sites of pericyte ablation. We show that the dye stays in circulation for several minutes after a single *i.v.* injection, providing more time for the dye to traverse the endothelium should the BBB be compromised. But, we still found no evidence of leakage in these studies (**Figure 5a-d**).

With respect to microglial responses, the study of Nikolakopoulou demonstrated that there was not an increased density of microglia after global pericyte ablation using the DTX strategy. However, these histological studies did not assess how the cells interacted with the capillary walls. Prior work from Lou and Nedergaard¹⁰ also showed that microglia extend juxtavascular processes in response to mild capillary injury with leakage, but no loss of flow. Studies of Davalos et al., show that microglia are very sensitive to BBB leak of fibrinogen.¹² Further, in a recent study from Vasquez-Liebana et al., focal BBB leakage caused by adult-induced pericyte depletion was associated with microglial reactivity. We therefore believe it is reasonable to treat microglia as sensitive surveillants of BBB leakage.

We aged pericyte-microglial double transgenic mice and performed additional experiments to examine microglial responses to pericyte loss; our prior submission only had experiments from adult mice. We find the same result that microglial reactivity is focused wholly on the soma of the ablated pericyte (site of laser irradiation) but not the capillary wall previously covered by processes of the ablated cell (**Figure 5e-j**). This provides further support that there lacks overt leakage of blood molecules (*i.e.*, fibrinogen) across the endothelium in these uncovered regions.

We have also added new data using pericyte-Claudin-eGFP double transgenic mice, to show that Claudin-rich tight junction strands were not overtly disrupted after pericyte ablation. In prior studies,

Claudin-eGFP in the endothelium showed aberrant loops, protrusions and gaps when disrupted by cerebral ischemia.¹³ We noticed transient changes in tight junction protrusions, suggesting endothelial remodeling, but not emergence of gaps or discontinuities in the tight junction strands as seen with paracellular BBB damage during stroke (**Supplementary Fig. 7**).

Finally, astrocytic endfeet are a component of the BBB, and we provide new data showing that the apposition of astrocyte endfeet remains intact following focal pericyte ablation (**Supplementary Fig. 8**).

Together, these data show that with the scale of pericyte loss produced here, there is not overt BBB leakage or damage to endothelial structure. This does not contradict prior work showing BBB leakage with more extensive pericyte loss using DTX ablation¹⁴ or other genetic pericyte-deficient models.⁴ It is likely that pericytes use paracrine signaling pathways to ensure endothelial health and ablation of a few cells does not affect overall barrier function. We now state this in the revised manuscript.

6. Please clarify what is meant by "transient" capillary stalls. For example, transient stalls were those detected only for a few minutes in a recent study (ref. 25). From figure 8d, it seems like the particular vessel noted experienced a series of stalling events, given the different locations of the RBC shadows in the vessel over different days.

Transient stalls in this context means that there was a lack of blood cell flow in the capillary segment on one day of imaging, but flow was again seen in a subsequent day of imaging. Our sampling intervals spanned days/weeks (day 0, 3, 7, 14, 21), so we unfortunately cannot report the duration of each stall observed. We have clarified this in the revision.

Minor:

7. Please clarify the sentence on lines 117-118: "...inducing endothelial contact with multiple pericytes."

We have reworded this paragraph for better clarity.

8. Figure 6a: The purple color is hard to see overlaid on the vessels.

We have improved visibility of this overlay. Please note that this is now **Figure 7a**, as we added an additional main figure.

9. Please provide a reference for the statement about the observation of string vessels in aging, Alzheimer's disease and VCID on lines 398-399.

We have added references for this statement.

10. Consider rephrasing the statement on lines 428-430 or providing additional citations to support it. Aberrant contraction of pericytes is a mechanism that can lead to blood flow impairment, but whether it is the "primary" mechanism is unclear.

We have removed this section during the process of honing the Discussion.

Reviewer #2 (Remarks to the Author):

The aetiology of age- and disease related vascular changes is an intriguing subject as these changes are likely central to the pathogenesis of a number of neurodegenerative disorders. Based on previous work by the same group, this study investigates the consequences of selective cortical pericyte loss in adult and aged mice, whereby the novelty is the comparison to the aged brain. Using a photo-thermal pericyte ablation, the authors perform a triple pericyte ablation in mice and examine the subsequent remodelling capacity of pericytes and its consequences for blood-brain barrier leakage, vascular tone and capillary flow comparing adult and aged mice. The study is well-performed, adequate methods are used and the manuscript is very well written.

The authors document a delayed and incomplete pericyte remodelling in aged mice resulting in reduced capillary tone. In both groups, pericyte loss leads to capillary flow abnormalities.

Major comments

1. Alterations in brain capillary functions are a well known phenomenon in the aging brain, however, the authors find no difference in cortical capillary structure in terms of capillary length, number of capillary junctions, number of pericyte somata (Figure 1 c, d and S1e) and in vascular tone between adult and aged mice.

-Was there any difference in pericyte (PDGFRB-tdTomato) density?

Thank you for this important point. Reviewer 1 made similar points in his/her comments 1 and 3, which we responded to above. In brief, we did not observe an age-related difference in capillary structure, pericyte density or capillary coverage by pericytes in the upper layers of cortex, where our in vivo studies were performed. This conclusion was based on rigorous re-analyses of our data in 3D.

-Were there age-related vascular changes in other brain regions, as this has been reported by other groups?

Our studies were focused on upper cortical layers and in the somatosensory cortex due to constraints of in vivo imaging and focal ablation. To maintain our focus on in vivo pericyte structural dynamics, we did not examine other brain regions by histology. However, prior studies have shown age-related loss in vascular density in different brain regions in mouse brain.^{1,2} Our data do not conflict with these findings, given the limiting sampling with our in vivo imaging. Indeed, in unpublished work from our lab, we find that capillary networks deeper in the brain (near and within the callosal white matter) have reduced density with age. This is a separate ongoing study using deep two-photon imaging that focuses on distinct topics from the current study.

2. It seems that the ability to extend processes varies highly (0-200 μm) between individual pericytes. In aged mice, more capillary surface remains uncovered at 21 days consistent with a lower number of pericyte-pericyte contacts. The authors conclude that the recruitment mechanism of pericytes is impaired in aged mice.

However, the data are incomplete to support this statement:

Despite the above statement, the authors show that there is no significance difference between groups for process length at day 21 (Figure 3 h), even though the processes have a significantly slower growth rate.

So why is there a larger capillary area that is still uncovered in aged mice? Was there a larger area uncovered after the lesion in the first place? Figure 3 I suggest that already at time point 0, both groups had very different starting positions in terms of uncovered vessel area. Could this be due to the choice of pericytes to be ablated: The authors state that the pericyte process length not only greatly varied between pericytes, but also determined their growth capacity.

We believe that differences in process length between age groups is masked the high variance in pericyte process growth within groups. Additionally, these data are a composite from pericytes of different zones, and zone-specific differences in aged mice may add to variance (**Figure 4**). This makes statistical detection of age-related differences challenging. While the graphs show non-significance, there is a trends toward lower extension achieved and total process length in aged mice (**Figure 3g,h**). The compounding of these differences make it such that capillary network fails to regain coverage after

21 days in the aged mice. We had therefore plotted **Figure 3i** to show the outcome of the remodeling efforts, which is the key takeaway from the data.

In the revision, we mention that statistical detection of any differences in these final metrics is challenging, because the variance of remodeling capacity is high between pericytes and the data is a composite of pericytes residing in different vascular zones that respond differently with aging (**Figure 4**).

In figure S3c they show that there was no difference in baseline length of pericyte processes between groups. However, figure S3c does not show the baseline length of the lesioned pericytes. If only 3 pericytes are depleted per animal, the choice is of course important if it determines their growth capacity. This data should be possible to retrieve from pre-lesion images and should be presented to control for any selection bias. Such differences in pericyte selection would also account for the larger uncovered area in aged mice at the starting point.

Regarding our ability to see how much of a “footprint” each pericyte had prior to ablation, due to immediate successive ablation of the three contiguous pericytes per experiment, there was not a point at which we could calculate the territories of individual pericytes in adult and aged mice. Thus, we cannot rule out the possibility that some pericytes chosen were inadvertently larger in the aged group. However, we do not think that selection bias was an issue in our studies, since there were limited opportunities to find three contiguous pericytes with ideally protruding somata for ablation.

The second determinant of pericyte growth is splitting which is dependent on the number of bifurcations in the depleted territory. The authors state that there was no difference in the number of bifurcations, but data supporting this are not presented.

This is a great point we had not considered. To address it we analyzed the number of bifurcations specifically within the regions uncovered by triple pericyte ablation between adult and aged mice. The analysis revealed that there was no significant difference (7.3 ± 2.7 and 7.9 ± 2.9 bifurcations in uncovered regions between adult and aged mice, respectively; $p=0.4035$, t-test; mean \pm sem). Thus, we do not think that impaired pericyte remodeling is because of reduced capillary connectivity in aged mice.

3. The authors state that they found no evidence of pericyte proliferation or cell recruitment in their paradigm: How was this assessed since the imaging interval is days to weeks?

Did the authors stain for cell division markers? How was pericyte recruitment excluded?

Use of constitutive Cre driver would allow for any new PDGFRbeta-positive cells appearing in the area to be labeled.¹⁵ Pericyte somata are quite easy to detect in the PDGFRbeta-tdTomato mice. All pericyte somata, besides the ones we ablated, were accounted for. We did not note the appearance of new pericytes in any experiment. In recent studies from developing brain using the same genetic labeling approach, we have seen what pericyte soma migration and proliferation looks like, and we are certain that this does not occur in the adult or aged cortex.¹⁶ We have added more detail on this in the revised manuscript.

4. Differences across microvascular zones

Here the authors compare 3 different anatomical regions in 2 different groups. Judging from Figure 4 c, aged mice extend longer processes along venules than adult mice. However, the graphs are separated by groups, whereas this was not done in Figure 4 f.

All conditions should be presented together and analysed by 2-way ANOVA.

We have now presented this information together and analyzed by two-way ANOVA as recommended, in Figure 4e,f.

5. Blood Brain Barrier (BBB) leakage is an important consequence of vascular alterations and seen as one of the implications of pericyte loss and a large contributor to neurodegeneration. In the current study paradigm, the authors inject only one dye of a large molecular weight and state that the pericyte loss was insufficient to cause BBB leakage. However, BBB leakage can be subtle, especially if the pericyte loss is so focussed. It would be extremely interesting to see whether any subtle BBB leakage can be reversed upon pericyte remodelling. I suggest the authors inject a smaller molecule and report the circulation time before imaging. Circulation time, details of the volume, concentration, supplier and average weight of the mice need to be reported for the dyes injected.

Thank you for raising this point, which was also brought up by Reviewer 1 (please see response to Comment 5 for Reviewer 1). As this reviewer brings up, the circulation time of the dye is quite important. Very low molecular weight dyes leave circulation in a matter of seconds, leaving little time to leak into the parenchyma. Thus, we chose a 10 kDa dye which we show stays in circulation for several minutes. These data are now in Figure 5a,b. We have also added details of the leakage experiment and mice in the Methods section.

Minor comments: Please correct sentences

Line 61: perfusion blood cells

Fixed.

Line 83: on how pericyte loss related in deficiencies

Fixed.

1. Thermal photo method for pericyte ablation. The researchers assumed that the focused photon on the pericyte soma causes the ablation of the pericyte. They exclude the possibility of photobleaching rather than killing pericyte cells, based on that the loss of pericyte signal was permanent while photobleaching is temporary. What is the time window of the temporary bleaching? Please support with references.

There are two reasons why we think that loss of pericyte fluorescence in our ablation method is not solely photobleaching. First, with photobleaching we would expect tdTomato fluorescence within the pericyte to be lost specifically where the region of laser irradiation occurred, at the soma. However, we find that tdTomato signal throughout the entire cell is lost, including the long processes of the pericyte that received no irradiation. Second, fluorescence recovery after photobleaching of cytosolic tdTomato can reach pre-bleach levels in neurons within 2 minutes.¹⁷ However, when we see this occur during post-ablation observation, we irradiate the cell again to ensure that it is fully ablated.

This suggests that pericytes die through necrotic injury. It is possible that some cells fail to die after ablation and later express more tdTomato to be seen at the 3 day imaging period. We provide an example of one of these failed experiments below (omitted from analysis). However, in most cases, no pericyte is seen at 3 days and the neighboring pericytes grow into the territory of the ablated cell. We unfortunately could not chart the time-course over which fluorescence returned in these failed experiments because imaging was not performed continuously. In the revised Methods, we now explain why this might not be photobleaching.

2. Quantification of the vascular length

The authors used the ImageJ (AngioTool) to estimate vessel length. How was the 90 um thick image processed? Is the analysis done in 3D or 2D? If 2D, how do they deal with possible overlapping between the vessels at different depths when converted to max projection.

Our original analysis was indeed performed in 2D, but we have now re-analyzed the data in 3D using Imaris software. Please refer to comment 1 from Reviewer 1, and **Supplementary Fig. 1c,d**.

Reviewer #3 (Remarks to the Author):

The manuscript presents a study of correlation between deficiency in pericyte remodeling and alterations in microvascular flow in an aging brain. Pericyte recovery is found to slow down with age in a mouse model. Furthermore, pericyte loss appears to result in local vasodilation of capillaries, which affects blood flow distribution. As a result, alterations in blood flow cause remodeling of microvascular structure, potentially leaving certain portions of the brain tissue underperfused. In addition, some capillaries may receive a little amount of erythrocytes due to blood flow alterations.

I think this is a very interesting hypothesis, which clearly deserves an investigation. The presented data support that there is a correlation between pericyte loss and the remodeling of microvasculature, however, it is not clear whether it would be the main cause or a contribution to the vascular remodeling. Therefore, I would be a bit careful with some of the statements. Below I provide a few suggestion which I hope will be useful for the authors to improve the manuscript.

1) The authors often use the word "maldistribution" referring to altered blood flow. I agree that blood flow is changed, but it is much less clear whether blood flow becomes kind of malfunctional or pathological.

Thank you for this important point. We suggest that the flow is maldistributed (or abnormal) for several reasons. First, we show that basal flow heterogeneity among capillaries is significantly increased by the focal capillary dilations occurring after pericyte loss (**Fig. 7d**). We have also performed additional analyses of in silico data to confirm that heterogeneity in affected regions is beyond the normal range seen in the capillary bed (**Fig. 8g,h**). A number of in vivo imaging studies have shown that flux homogenization among capillaries occurs during functional hyperemia.¹⁸⁻²¹ This homogenization from a basal heterogeneous facilitates oxygen extraction²², and empirical studies show a relationship between more homogenous capillary flux and tissue oxygen content.²³ By increasing flow heterogeneity above basal levels, we increase the threshold to achieve flux homogenization. This would disrupt the delivery of oxygen to brain tissue on a local scale and could make flow regulation more difficult.

Second, we show that stalling of capillary flow becomes a regular observation in regions of pericyte loss (64% of pericyte ablation experiments). In these regions, we estimate that an average of 3%, and maximum of 9%, of capillaries in the region (including dilated vessels, Gen1 and Gen2 neighbors) would experience a stall in flow. Stalling of capillary flow in the healthy adult brain is typically sparse, affecting only 0.4% of capillaries in a prior study by the Schaffer and Nishimura labs.²⁴ Increased capillary stalling to only 1.8% of total capillaries in a mouse model of Alzheimer's disease was associated with worsened cognitive performance.²⁴

Third, the observation of capillary regressions is also a very rare event in the adult brain, and prior longitudinal imaging studies have shown marked stability of capillary structure.²⁵ Yet, we observe regressions in 13% of the stalled capillaries. Thus, we feel that if these types of changes were occurring broadly throughout the capillary network, as expected with extensive pericyte loss, there would be significant perturbations to capillary flow and structure.

However, we recognize the limitations of our studies, which do not show definitively that the capillary flow changes have altered tissue oxygen levels or neuronal function. We have added to the Discussion, the need for additional studies to study the tissue consequence of focal pericyte loss using novel oxygen imaging probes.

2) The blood steal effect is called the Zweifach-Fung effect, which states that at bifurcations, daughter branches with a larger flow rate generally receive a larger erythrocyte content. This needs to be properly explained and cited.

Thank you for this suggestion. We have revised the text accordingly to explain the Zweifach-Fung effect, and to cite the relevant paper, in both the Results and Methods sections.

3) The presented data show that capillary regression is strongly associated with "capillary stalls" (no significant flow). However, no difference in stall or regression frequency was found between different age groups. Since your proposition of microvascular remodeling is mainly associated with "capillary stalls", then how would it occur differently for different ages?

Due to the focal nature of our manipulations, we lacked the statistical power to compare stalls and regressions between ages. Our *in silico* studies suggested that greater dilations seen with aging would cause greater capillary flow heterogeneity and greater flow steal, which should in turn lead to higher likelihood for stalling and regressions. In new analyses, we tried to quantify stalling in the *in silico* data (please see comment 6 below). But the results pointed to other biological reasons for capillary stalls, not captured by the modeling approach. While this specific question is difficult to address with our focal ablation approach, a recent study using global pericyte ablation reported increased capillary stalling, in support of our conclusions.²⁶

4) Does brain microvasculature have also functional vasodilation/vasoconstriction, which generally takes place at the arteriolar (pre-capillary) part of microvasculature? How do you exclude such effects?

This is a great question, considering studies showing that electrical propagation of signals from capillaries can reach upstream arterioles.²⁷ From our images (**Fig. 4b, Supplementary Fig. 5a,b, 13b**), we do not see major alterations in the basal diameter of the arteriole-capillary transition zone after pericyte ablation. As the reviewer notes, however, the question is also relevant to vasoreactivity in transitional vessels during functional hyperemia, and the effects of capillary pericyte loss. This is something we are carefully analyzing in ongoing studies with awake mice, and is beyond the scope of our focus on capillary network dynamics.

5) Would angiogenesis kick in if some portions of brain tissue are becoming hypoxic? Or would it be irrelevant for some reason?

This is a very interesting question. We have experience detecting developmental angiogenesis in cortex using pericyte-labeled mice and intraluminal dyes.¹⁶ We did not observe any new vessel growth

at any time point post-ablation in our focal ablation experiments. In a prior study, we reported one instance of new vessel growth.²⁸ Thus, angiogenesis can occur but is very rare with the scale of pericyte loss induced here. There is decreased signaling through the major pathway to promote angiogenesis, VEGF-VEGFR, with aging. A recent study showed that this is due to increased VEGF sequestration by the natural decoy receptor for VEGFR, sFlt1.²⁹ The chances of local hypoxia inducing angiogenesis may therefore decrease with aging.

6) I think the main difficulty in simulations is to assign proper boundary conditions for numerous inlets and outlets. How do you do this? Do you make any 'semiquantitative' comparison with experiments? How sensitive and robust your simulation results to the choices of boundary conditions made? Do you see any capillary stalls in simulations?

Indeed, assigning suitable boundary conditions is one of the difficulties in performing simulations in realistic microvascular networks. The approach we used and the comparison to in vivo data is described in one of our previous works (see Figure 1 and Table 3 of Schmid *et al.* 2017³⁰).

In brief, the realistic microvascular network is implanted in the center of a large semi-realistic network (realistic penetrating trees + artificial capillary bed). The resulting compound network has only arteriole and venule in- and outflows located at the cortical surface. We assign diameter-dependent pressure values based on experimental data from literature. Subsequently, we compute the pressure field in the large compound network for a constant hematocrit and assign the resulting pressure values as boundary conditions at the in- and outflows of the realistic network. These pressure boundary conditions are kept constant for the simulations with RBC tracking.

To validate our simulation results we compared the resulting flow field to in vivo measurements of RBC velocity and RBC flux in descending arterioles and capillaries. As the simulated characteristics agree well with data from literature, we are confident that our pressure boundary conditions approach allows us to simulate a physiological microvascular flow field.

Our previous investigations showed that the sensitivity with respect to boundary conditions is low. The assigned pressure boundary conditions at the capillary level mostly affect the vessels directly at the boundary (~2-3 segments). To avoid effects from boundary conditions we focus our analysis on areas in the center of microvascular network and excluded all capillaries directly adjacent to the boundary from the analysis.

Regarding stalls, we can detect stalled/low flow capillaries in the simulations. In MVN1 and MVN2, 5% of all capillaries in the upper 200 μm have very low RBC velocity <0.04 mm/s and <0.1 mm/s, respectively, that might be similar to a stalled capillary in vivo. We used the 5th percentile as a threshold for stalled/low flow capillaries to study if their frequency increases in response to varying degrees of capillary dilation seen in adult and aged mice (**Supplementary Figure 12d-e**). However, these analyses did not reveal greater numbers of stalls. In the simulations, stalling would be a direct consequence of the flow and pressure field, and this outcome points to additional biological factors (e.g. leukocyte adhesion) that are not accounted for in the numerical model.

7) I think it would be interesting to look at an effective 'oxygen delivery' in simulations, which would be a product of local erythrocyte content and flow. For example, your model implements bifurcation laws (or the Zweifach-Fung effect), so that erythrocyte content can be locally assessed. Do you observe significant erythrocyte steals at some bifurcations? How strong such changes can be?

We indeed see strong evidence for blood flow steal at capillary bifurcations experiencing uneven dilation in our in silico studies (**Fig. 8f**). As suggested by the reviewer the blood flow steal also affects the RBC distribution (Zweifach-Fung effect). Consequently, the relative RBC flux changes in the dilated outflow branch are larger than the relative flow changes (compare **Fig. 8f** and **Supplementary Fig. 12a**, median flow changes in dilated outflow branch for increasing diameter changes: 54%, 86%, 118%, median RBC flux changes: 57%, 116%, 186%).

We agree that oxygen transport simulation would be a valuable and interesting for studies on the effects of pericyte loss. As flow alterations caused by focal pericyte ablation occur on the local scale, the employed oxygen transport model should ideally be able to resolve oxygen transport by individual RBCs and within realistic vascular topologies.³¹ However, such simulations are computationally expensive and require careful analysis of results and sensitivities. Consequently, they are beyond the scope of the current manuscript. Just as importantly, we plan to use in vivo oxygen imaging probes³² to directly visualize tissue oxygen content during pericyte ablation. These experiments are challenging and a work in progress.

References

1. Schager, B. & Brown, C.E. Susceptibility to capillary plugging can predict brain region specific vessel loss with aging. *Journal of Cerebral Blood Flow & Metabolism* **40**, 2475-2490 (2020).
2. Murugesan, N., Demarest, T.G., Madri, J.A. & Pachter, J.S. Brain regional angiogenic potential at the neurovascular unit during normal aging. *Neurobiology of Aging* **33**, e1-16 (2012).
3. Soto, I., *et al.* APOE Stabilization by Exercise Prevents Aging Neurovascular Dysfunction and Complement Induction. *Plos Biology* **13**, e1002279 (2015).
4. Vazquez-Liebanas, E., *et al.* Adult-induced genetic ablation distinguishes PDGFB roles in blood-brain barrier maintenance and development. *Journal of Cerebral Blood Flow & Metabolism* **2**, 264-279 (2021).
5. Wu, Y.-T., *et al.* The cellular architecture of microvessels, pericytes and neuronal cell types in organizing regional brain energy homeostasis in mice. *bioRxiv* <https://doi.org/10.1101/2021.05.19.444854> (2021).
6. Reeson, P., Choi, K. & Brown, C.E. VEGF signaling regulates the fate of obstructed capillaries in mouse cortex. *Elife* **7** (2018).
7. Montagne, A., *et al.* Pericyte degeneration causes white matter dysfunction in the mouse central nervous system. *Nature Medicine* **24**, 326-337 (2018).
8. Hartmann, D.A., Coelho-Santos, V. & Shih, A.Y. Pericyte control of blood flow across microvascular zones in the central nervous system. *Annual Review of Physiology* **84**, 331-354 (2021).
9. Vanlandewijck, M., *et al.* A molecular atlas of cell types and zonation in the brain vasculature. *Nature* **554**, 475-480 (2018).
10. Lou, N., *et al.* Purinergic receptor P2RY12-dependent microglial closure of the injured blood-brain barrier. *Proceedings of the National Academy of Sciences* **113**, 1074-1079 (2016).
11. Armulik, A., *et al.* Pericytes regulate the blood-brain barrier. *Nature* **468**, 557-561 (2010).
12. Davalos, D., *et al.* Fibrinogen-induced perivascular microglial clustering is required for the development of axonal damage in neuroinflammation. *Nature Communications* **3**, 1227 (2012).
13. Knowland, D., *et al.* Stepwise recruitment of transcellular and paracellular pathways underlies blood-brain barrier breakdown in stroke. *Neuron* **82**, 603-617 (2014).
14. Nikolakopoulou, A.M., *et al.* Pericyte loss leads to circulatory failure and pleiotrophin depletion causing neuron loss. *Nature Neuroscience* **22**, 1089-1098 (2019).
15. Hartmann, D.A., *et al.* Pericyte structure and distribution in the cerebral cortex revealed by high-resolution imaging of transgenic mice. *Neurophotonics*, 041402 (2015).

16. Coelho-Santos, V., Berthiaume, A.A., Ornelas, S., Stuhlmann, H. & Shih, A.Y. Imaging the construction of capillary networks in the neonatal mouse brain. *Proc Natl Acad Sci U S A* **118**, e2100866118 (2021).
17. Makino, H. & Malinow, R. AMPA receptor incorporation into synapses during LTP: the role of lateral movement and exocytosis. *Neuron* **64**, 381-390 (2009).
18. Lee, J., Wu, W. & Boas, D.A. Early capillary flux homogenization in response to neural activation. *Journal of Cerebral Blood Flow & Metabolism* **36**, 375-380 (2016).
19. Li, Y., Wei, W. & Wang, R.K. Capillary flow homogenization during functional activation revealed by optical coherence tomography angiography based capillary velocimetry. *Scientific Reports* **8**, 4107 (2018).
20. Stefanovic, B., *et al.* Functional reactivity of cerebral capillaries. *Journal of Cerebral Blood Flow & Metabolism* **28**, 961-972 (2007).
21. Gutiérrez-Jiménez, E., *et al.* Effect of electrical forepaw stimulation on capillary transit-time heterogeneity (CTH). *Journal of Cerebral Blood Flow & Metabolism* **36**, 2072-2086 (2016).
22. Jespersen, S.N. & Østergaard, L. The roles of cerebral blood flow, capillary transit time heterogeneity, and oxygen tension in brain oxygenation and metabolism. *Journal of Cerebral Blood Flow & Metabolism* **32**, 264-277 (2012).
23. Li, B., *et al.* More homogeneous capillary flow and oxygenation in deeper cortical layers correlate with increased oxygen extraction. *Elife* **8**, e42299 (2019).
24. Cruz Hernández, J.C., *et al.* Neutrophil adhesion in brain capillaries reduces cortical blood flow and impairs memory function in Alzheimer's disease mouse models. *Nature Neuroscience* **22**, 413-420 (2019).
25. Cudmore, R.H., Dougherty, S.E. & Linden, D.J. Cerebral vascular structure in the motor cortex of adult mice is stable and is not altered by voluntary exercise. *Journal of Cerebral Blood Flow & Metabolism* **37**, 3725-3743 (2017).
26. Choe, Y.G., *et al.* Pericyte Loss Leads to Capillary Stalling Through Increased Leukocyte-Endothelial Cell Interaction in the Brain. *Frontiers in Cellular Neuroscience* **Epub ahead of print** (2022).
27. Longden, T.A., *et al.* Capillary K⁺-sensing initiates retrograde hyperpolarization to increase local cerebral blood flow. *Nature Neuroscience* **20**, 717-726 (2017).
28. Berthiaume, A.A., *et al.* Dynamic remodeling of pericytes in vivo maintains capillary coverage in the adult mouse brain. *Cell Reports* **22**, 8-16 (2018).
29. Grunewald, M., *et al.* Counteracting age-related VEGF signaling insufficiency promotes healthy aging and extends life span. *Science* **373**, eabc8479 (2021).
30. Schmid, F., Tsai, P.S., Kleinfeld, D., Jenny, P. & Weber, B. Depth-dependent flow and pressure characteristics in cortical microvascular networks. *PLoS Computational Biology* **13**, e1005392 (2017).
31. Lückner, A., Secomb, T.W., Barrett, M.J.P., Weber, B. & Jenny, P. The Relation Between Capillary Transit Times and Hemoglobin Saturation Heterogeneity. Part 2: Capillary Networks. *Frontiers in Physiology* **9** (2018).
32. Esipova, T.V., *et al.* Oxyphor 2P: A high-performance probe for deep-tissue longitudinal oxygen imaging. *Cell Metabolism* **S1550-4131**, 30759-30759 (2019).

REVIEWERS' COMMENTS

Reviewer #1 (Remarks to the Author):

The authors have satisfactorily addressed the comments. No further comments.

Reviewer #2 (Remarks to the Author):

The authors have responded to all points raised by me and addressed them sufficiently.

Reviewer #3 (Remarks to the Author):

The authors have addressed satisfactorily all my comments, so that I have no further suggestions and recommend the manuscript for publication.